

# Analysis fire patterns and drivers with a global SEVER-FIRE model incorporated into Dynamic Global Vegetation Model and satellite and on-ground observations.

Sergey Venevsky[1], Yannick Le Page[2], José M. C. Pereira[2], Chao Wu[1]

1 Ministry of Education Key Laboratory for Earth System Modeling, Department of Earth System Science, Tsinghua University, Beijing 100084, China

Technical University of Lisbon, Instituto Superior de Agronomia, Department of Forestry, Tapada da Ajuda 1349-017 Lisboa, Portugal

*Correspondence to*: Chao Wu(wuc14@mails.tsinghua.edu.cn) and Sergey Venevsky (venevsky@tsinghua.edu.cn)

**Abstract.** Biomass burning is an important environmental process with a strong influence on vegetation and on the atmospheric composition. It competes with microbes and herbivores to convert biomass to $CO_2$ and it is a major contributor of gases and aerosols to the atmosphere. To better understand and predict global fire occurrence, fire models have been developed and coupled to Dynamic Global Vegetation Models (DGVMs) and Earth System Models (ESMs).

We present SEVER-FIRE (Socio-Economic and natural Vegetation ExpeRimental global fire model which is incorporated into the SEVER-DGVM. One of the major focuses of SEVER-FIRE model is an implementation of pyrogenic behaviour of humans (timing of their activities and their willingness/necessity to ignite or supress fire), related to socio-economic and demographic conditions in a geographical domain of the model application. Burned areas and emissions from the SEVER model are compared to the Global Fire Emission Database version 2 (GFED), derived from satellite observations, while number of fires are compared with regional historical fire statistics. We focus both on the model output accuracy and on its assumptions regarding fire drivers, and perform:

1- An evaluation of the predicted spatial and temporal patterns, focusing on fire incidence, seasonality and inter-annual variability.

2- Analyses to evaluate the assumptions concerning the etiology, or causation, of fire, including climatic and anthropogenic drivers, as well as the type and amount of vegetation.



SEVER reproduces the main features of climate driven inter-annual fire variability at a regional scale, such as the large fires associated with the 1997-98 El Niño event in Indonesia, Central and South America, which had critical ecological and atmospheric impacts. Spatial and seasonal patterns of fire incidence reveal some model inaccuracies, and we discuss the implications of the distribution of vegetation types inferred by the DGVM, and of assumed proxies of human fire practices. We further suggest possible development directions, to enable such models to better project future fire activity.

## 1 Introduction

The biosphere is affected by fires through physical and chemical pathways, involving interactions between the terrestrial and atmospheric components of carbon, water and nutrients cycles. As a natural phenomenon, fires are an integral part of a majority of ecosystems, influencing soil fertility, stand regeneration, vegetation composition, and succession (Le Page et al., 2015; Levine et al., 1999). However, through its anthropogenic use for land management (agriculture, pasture, deforestation), fire incidence is considerably higher than under natural conditions in many regions, including savannas in Africa and Australia, or tropical forests in South America and South East Asia (Bond et al., 2005).

Abundant literature points a variety of impacts, roles, and drivers of fires, and an extended range of spatial and time scales involved. It is estimated that, on average, an area equivalent to that of India burns every year, predominantly in savannas and grasslands (Tansey et al., 2004). Burned areas in tropical and boreal forests are smaller, but their high productivity and carbon storage capacity results in significant emissions of numerous greenhouse gases (e.g. $CO_2$, $CH_4$, (Andreae and Merlet, 2001); (Pereira et al., 1999)). Globally, total fire emissions are equivalent to approximately one third of fossil fuel burning emissions (Le Quéré et al., 2015; van der Werf et al., 2006b; Wu et al., 2017). Net emissions, stemming from deforestation or increased fire activity, are much smaller, but poorly constrained (van der Werf et al., 2006a), and highly variable on inter-annual time scales, especially through induced changes in fire sensitivity of highly productive ecosystems by El Niño/La Niña and other climatic phenomena (Duncan et al., 2003; Langenfelds et al., 2002; Le Page et al., 2008; van der Werf et al., 2004; van der Werf et al., 2008).

The strong integration of fires with the biosphere system is also emphasized by their dependence on a complex system of interactive drivers, designated as the fire triangle (Schoennagel et al., 2004), dominated by climate, vegetation and human activities. Precipitation rates and temperature partly control the amount of fuel available to burn, its moisture content, and fire behaviour in case of ignition (Crevoisier et al., 2007; Turner et al., 2008). Fire incidence, fire severity, and ensuing emissions are also dependent





on the vegetation types, structure and productivity of the ecosystem (Andreae and Merlet, 2001; Hammill and Bradstock, 2006). Finally, anthropogenic activities, as mentioned above, greatly bias the natural occurrence of fires, increased in many regions as a land management tool, or decreased through fire suppression strategies (firefighting, preventive fires, (Veblen et al., 2000)). Other factors are involved

(topography, natural landscape breaks, grazing), but most important is the interaction between those drivers, which needs to be considered to yield relevant information about fire risk (Dwyer et al., 2000a).

Dynamic Global Vegetation Models (DGVMs) and Earth System Models (ESMs) simulate vegetation dynamics at global scale, fire is included as an explicit process in only a few of these models (Arora and Boer, 2005; Bachelet et al., 2001; Li et al., 2013; Thonicke et al., 2010; Thonicke et al., 2001; Venevsky

et al., 2002; Wu et al., 2017; Yue et al., 2014). Given the importance of fires and their dependence on various model inputs or simulated processes, the development of fire modules is of great interest to understand and evaluate the fire related couplings and feedbacks assumptions. Some global fire models are designed to study impact of anthropogenic phenomena in global fire dynamics and are not incorporated into DGVMs or ESMs (Le Page et al., 2015), but rather give further insight for fire modelling

within Earth and biosphere simulators. Last global fire models designed for DGVMs (Li et al., 2013; Thonicke et al., 2010) contain sets of rather complicated equations with variety of coefficients (despite they name themselves intermediate complexity models) which is hard to obtain unless satellite derived functions are used. Some of global fire models use satellite observations as input (e.g SPITFIRE (Thonicke et al., 2010) for number of lightning strikes). Such features hinder further use of these global

fire models for climate change and socio-economic change studies, which should be based solely on understandable physical reasoning. Assessment against observation data of global fire models within DGVMs is now based mainly on satellite data. This limits validation to only one registered quantitative characteristics of global fire regime, namely area burnt (another satellite based variable global carbon emission is as a rule result of a model itself) and at a limited time interval (as a rule last two-three decades).

Meanwhile, historical fire statistics exists in all major forested countries and includes numbers of registered fires by case (lightning or human) and areas burnt starting at least from sixties of the last century. Ignoring of historical statistics leads to visible shortcoming in description of regional fire regimes. For example, LPJ-DGVM based SPITEFIRE global fire model (Thonicke et al., 2010), which was historically follow up of Reg-FIRM regional LPJ-DGVM fire model (Venevsky et al., 2002) used for simulation

numbers of fires and areas burnt in Spain, overestimates number of fires (see Figure 3 c in SPITFIRE (Thonicke et al., 2010)) in Iberian Peninsula three to four times. SEVER-FIRE (Socio-Economic and natural Vegetation ExpeRimental global fire model is incorporated into the SEVER_DGVM (Venevsky and Maksyutov, 2007; Wu et al., 2017), which is a modification of LPJ-DGVM (Sitch et al., 2003) for




daily time step computation. SEVER-FIRE model is a follow up of Reg-FIRM and is designed using principles of the last. No satellite derived data are used as an input of the model. Only physically based or just 'common sense' based equations from on-ground observations allow direct implementation of SEVER-FIRE model in any DGVM or ESM for investigation of future global change impacts or past

global fire regimes reconstruction. Unlike in other global DGVM fire modules (Li et al., 2013; Thonicke et al., 2010) all equations are kept simple following ideology of Reg-FIRM. One of the major focuses of SEVER-FIRE model is an implementation of pyrogenic behaviour of humans (timing of their activities and their willingness/necessity to ignite or supress fire), related to socio-economic and demographic conditions in a geographical domain of the model application. Importance of description of pyrogenic

behaviour of humans are confirmed by recent findings of bi-modal fire regimes, reflecting human fingerprint in global fires dynamics (Benali et al., 2017), as well as by differences in timing of ignitions determined by religious background in Sub-Sahara Africa (Pereira et al., 2015). Fire weather regimes, set by climate dynamics, and fuel state set by vegetation dynamics are other important drivers in SEVER-FIRE model. SEVER-DGVM fire module, based on climate observations, external anthropogenic

parameters, and SEVER-DGVM derived vegetation, estimates fire incidence and emissions. The resulting vegetation disturbance feeds back to the DGVM, ensuring a fully coupled system (see model description).

We perform a comparison of SEVER outputs with fire data derived from satellite sources, the Global Fire Emission Database version 2 (GFED) (van der Werf et al., 2006a), as well as with historic fire data with two objectives. First, a global evaluation of a DGVM-fire model, focusing on crucial and simple features,

namely fire incidence, seasonality, inter-annual variability, and emissions. Second, (the most important) by identifying the reasons for large inconsistencies we propose further modifications to SEVER-FIRE. As it was already mentioned, current global fire modules feature conceptual differences, but are generally based on similar assumptions. Thus, this study may provide relevant information for improvement of other current DGVM fire modules. The work presented in this paper is partly based on the Ph. D. thesis

by Y. Le Page. We therefore signal the reader that significant parts of the text in the sections 3 and 4 already appeared in Le Page (2009).

## 2. Data and Methods

### 2.1 SEVER-DGVM and SEVER-FIRE Models

#### 2.1.1 Input of DGVM to fire model



SEVER-DGVM is a coupled vegetation-fire mechanistic model designed to run at a range of temporal (daily to monthly) and spatial (10 km to 2.5 ° with 0.5º mostly tested ) resolution levels (Venevsky and Maksyutov, 2007). The fire module SEVER-FIRE is a further development of the Reg-FIRM (Venevsky et al., 2002), which was applied only for Iberian Peninsula, from a regional to the global scale. Unlike the other global fire models (Thonicke et al., 2010), which use conceptual approach of Reg-FIRM for design of process-oriented fire model, SEVER-FIRE model does not include satellite based derived relationships, but only equation based on field/laboratory observations and "common sense" hypothesis, when on-ground data are not available. The aim of this model is to provide at the global scale a fully mechanistic description of major characteristics registered in standard fires statistics and/or satellite observations around the world, namely number of fires, area burnt and carbon emissions. An important goal of SEVER-FIRE model is inclusion in Earth System models (Bonan and Doney, 2018; Bowman et al., 2009) in order to make realistic climate change predictions of global wildfire dynamics The most important variables, provided by SEVER-DGVM for SEVER-FIRE model include the distribution of 10 Plant Functional Types , which are similar to LPJ-DGVM vegetation types (see names of PFT in Table 1) over the globe, described as a distribution of fractions within a grid cell $Cveg_{pft}$, net primary productivity $NPP_{pft}$, carbon of aboveground vegetation $c_{pft}$, fuel loading $lit_{pft}$, described as a mass of litter, and soil moisture in the upper 0.1 m layer$m$ (see Table 2 for description of fire model variables and parameters)

### 2.1.2 External input to fire model

Gridded climate, demographic and socio-economic data comprise external input for the fire module. Minimum/maximum daily temperature $t_{min/max}$, daily precipitation/convective precipitation $prec/cprec$ and wind speed $u$ are the climate variables used in SEVER-FIRE. Human population density, ratio of rural to total population (rural and urban population) $rur = \frac{P_{rur}}{P_{tot}}$, wealth index $WI$ and average distance from megacities $dist$ (recalculated with simplified assumptions from population density and ratio of rural to urban population) comprise socio-economic input to the fire model.

### 2.1.3 Output of fire model

The model separates human-induced (indexed as *hum*) and lightning fires (indexed as *nat*) by sources of ignition and all output variables of fire models can be obtained either by these two classes of fires of for both classes in total as their sum (not indexed). We omit the mentioned indexes in description of output variables further on for simplicity. The output of the model includes number of fires $Nfire$ , area burnt *aburnt* fire carbon emission $c\,fire$ , number of PFT's individuals killed $Nind_{pft}$ and updated vegetation





carbon and NPP. Fire model feedbacks to the DGVM through the increased area (equal to burnt areas by PFTs) and decreased number of PFT's individuals for competitive occupation by PFTs after a fire and updated carbon fluxes and pools for carbon cycle simulation within vegetation model.

Thus, the DGVM and fire module work in interactive mode, incorporating a representation of fire-vegetation feedbacks.

### 2.1.4 Components of SEVER-FIRE

The SEVER-FIRE model consists of six related components described below:

- Estimation of fire weather danger index and fire probability,

- Simulation of lightning ignition events and number of lightning fires,

- Simulation of human ignition events and number of human fires,

- Simulation of fire spread after ignition,

- Fire termination,

- Estimation of fire effects (areas burnt, pyrogenic emissions, number of each PFT individuals killed).

All six components are controlled by PFT dependent fire parameters (see list in Table 2)

#### 1) *Estimation of fire weather danger index and fire probability*

Fire weather danger index *FDI(d)*, measured from 0 ("no fire danger"), to 1 ("extreme fire danger"), is estimated in SEVER-FIRE based on the Reg-FIRM fire danger index (Venevsky et al., 2002). It is calculated at a daily time step as a multiple of exponentially normalized Nesterov Index (based on accumulated difference of minimum and maximum temperature, forced to zero by 3 mm daily precipitation threshold) and vegetation and soil moisture dependent fire probability. Using of Reg-FIRM based fire weather danger indexes, became popular in contemporary global fire modelling (Arora and Boer, 2005; Thonicke et al., 2010) mainly due to calculation simplicity. Direct comparison of fire risk for Siberia, described by more sophisticated Canadian Fire Danger and Russian Fire Danger Indexes (used by national Forest Service) in both countries with Reg-FIRM Fire Danger Index, revealed that they are almost their equivalent (Rubtsov, personal communication). The fire probability function is designed as a regression  from observations (Thonicke et al., 2001). It depends on current soil moisture in the upper 10 cm layer and PFT dependent fire moisture of extinction (Table 1), adapted from experimental study of Albini (1976).



### 2) Simulation of lightning ignition events and number of lightning fires

The number of potential lightning ignitions in a grid cell is calculated from the daily number of cloud-to-ground flashes $N_{flashes}$, which is estimated from convective precipitation as a non-linear regression polynomial function of power four (as in Allen Dale and Pickering Kenneth (2002)). Using of power four

polynomial function by convective precipitation to represent number of flashes has theoretical physical grounds (van Vonnegut, 1969). Allen and Pickering (2002) prepared their parametrisation of number of flashes for North America, so we made a validation test of $N_{flashes}$ for the globe (Venevsky, 2014) using OTD-LIS observed lightning data (Christian Hugh et al., 2003) and found that the parametrization performs well at global scale ($R^2$=0.51). Cloud-to-ground flashes are divided to negatively charged (90%)

and positively charged (10%) (Latham and Shielter, 1989). Only the flashes with long continuous current (LCC flashes, 75% of positively charged and 25% of negatively charged) can ignite wildfire (Latham and Shielter, 1989). Efficiency of LCC flashes to ignite depends from bulk density of fuel as it was shown in laboratory (Latham and Shielter, 1989), so number of efficient to ignite positive flashes $N_{flashes}{}_{pos}^{eff}$ and number of efficient to ignite negative flashes $N_{flashes}{}_{neg}^{eff}$ at first glance can be simplified as Eq. (1) and

Eq. (2):

$$N_{flashes}{}_{pos}^{eff} = N_{flashes} * 0.1 * 0.75 * b * a * dens * \bar{t}_{thunder}, \tag{1}$$

$$N_{flashes}{}_{neg}^{eff} = N_{flashes} * 0.9 * 0.25 * b * a * dens * \bar{t}_{thunder}, \tag{2}$$

where b =0.01 is efficiency of lightning t ignite (Latham and Shielter, 1989), a=0.25 m$^2$/kgC regression coefficient (simplified from Latham and Shielter, 1989), $\bar{t}_{thunder}$ is an average daily time of thunder over

a grid cell, set to one hour (Uman, 1987) and *dens* is bulk density of fuel (kgC/m$^2$). Bulk density of fuel is an important variable of SEVER-FIRE model, used in several basic equations. We assume that all PFTs found in a grid cell are distributed homogeneously and bulk density of fuel in a grid cell is calculated as Eq. (3):

$$dens = \sum_{i=1}^{Npft} Cveg_{PFT}(i) * dens_{PFT}(i), \tag{3}$$

where $Cveg_{PFT}(i)$ is foliar projection cover of *i*-th PFT, $dens_{PFT}(i)$ is bulk density of *i*-th PFT (see Table 1), which are taken from from Reg-FIRM (Venevsky et al., 2002) and study of Albini (1976), $Npft$ is total number of PFTs in a grid cell. Bulk density of fuel in the grid cell and depth of fuel (in cm), calculated as Eq. (4):





$$depth = 0.1 * \sum_{i=1}^{Npft} lit_{PFT}(i) \, / dens, \tag{4}$$

and they are translated into arriving daily number of natural ignitions from positive $N_{ignitions_{pos}}$ and negative flashes $N_{ignitions_{neg}}$, using generalisation in two functions of probaility to ignite for positive and negative flashes by eight fuel types of Anderson (2002), obtained from laboratory experiments (Eq. (5) and (6)):

$$N_{ignitions_{pos}} = (1/(1 + \exp(5.5 * (1./1.5) ** (((16. - dens)/16.) * 5) * 1.25 - 1.2 * 0.5 ** ((16. - dens)/$$
$$16. * 5. + 0.1) * depth))) * N_{flashes_{pos}}^{eff}, \tag{5}$$

$$N_{ignitions_{neg}} = (1/(1 + \exp(5.5 * (1./1.5) ** (((16. - dens)/16.) * 5) - 1.2 * 0.5 ** ((16. - dens)/16. *$$
$$5.) * depth))) * N_{flashes_{neg}}^{eff}. \tag{6}$$

Total number of arriving ignitions from effective positive and negative LCC flashes are recalculated into number of surving natural fires $N_{fires_{nat}}$ in a grid cell with area $S_{grid}$, which depends on daily fire danger index $FDI(d)$ maximum rate of surviving ignitions $rate_{survival_{max}}$, taken as 0.15 (Anderson, 2002) and soil moisture in 1 cm fuel layer, simplified as 10% of soil moisture *moist* in upper 10 cm, see Eq. (7):

$$N_{fires_{nat}}(d) = FDI(d) * (N_{ignitions_{pos}} + N_{ignitions_{neg}}) * rate_{survival_{max}} * (1 - moist * 0.1) *$$
$$S_{grid}. \tag{7}$$

Module of number of lightning fires was validated using data for lightning and lightning fires in central cordilierra of Canada (Wierzchowski et al., 2002). This study contains data for number of lightning fires for the years 1961-1994 and annual number of lightning strikes for 1989-1994 for the central cordilierra area 50-54°N, 114-120°W. The central mountain range in the area divides it into two parts, one is in British Columbia, another in Alberta provinces. SEVER-FIRE is able to reproduce values for total annual number of lightning strikes for both provinces (see Fig. 1) and number of lightning fires in the provinvces (see Fig. 2)..

The model reproduces three to two fold dominance of annual number of lighning strikes in Alberta and seven to ten fold dominance of annual number of lightning fires in British Columbia. Using of convective precipitation as a driver for number of lightning fires also confirmed by study of (Cardoso Manoel et al.,





2007), who found that lightning fire occurrence in Brazil is related to zonal flux of moisture in the atmosphere.

### 3) *Simulation of human ignition events and number of human fires*

The number of potential human ignitions $N_{ignitions\_human}(d)$ is calculated as a power function from population density with saturation, suggested by the Russian Forest Service and also used in the Reg-FIRM (Venevsky et al., 2002) multiplied by normalized socio-economic characteristics of population and by fuel conditions (similar to lightning ignitions) in the grid cell (see Eq. (8)):

$$(N_{ignitions\_human_j}(d) = 6.8 * P^{0.43} * \bar{a} * rate_{pop_j} * timing_j(d) * a * dens, \tag{8}$$

where $P$ is population density in persons per km$^2$, $\bar{a}$ is a mathematical expectation of number of ignition produced by one person for millions of hectares multiplied by $10^{-4}$ (coefficient a) to scale for square kilometers, $j$ is either rural ($j=rur$) or urban population ($j=urb$), $rate\_pop_j$ is a ratio of rural tp urban population, so that $rate\_pop_{rur} + rate\_pop_{urb} = 1$, $timing_j(d)$ is daily timing of pyrogenic activity of population. Timing of human pyrogenic activity $timing_j(d)$ at a first glance is defined separately for the northern and southern hemisphere and for rural and urban population as a step function, and it is mostly based on agricultural and vacation calendars (for city inhabitants). So, for example, for the entire northern hemisphere it was set to one in July, August (Summer vacations) for urban population, March, April, May (Spring agriculture activities) and September, October, November (Autumn agriculture activities) for rural population and to 0.5 in the rest of a year.

Mathematical expectation of number of ignition produced by one person for millions of hectares $\bar{a}$ is set to be an exponential function of wealth index WI, determined from the data of UN Human Settlemnet Program (see Eq. (9)):

$$\bar{a} = \exp(-7.65 * 10^{-2} * WI). \tag{9}$$

This assumes that maximum mathematical expectation of number of ignition produced by one person is eqaul to one for millions of hectares for a grid cell with the most theoretically possible poorest population (WI=0) and $\bar{a}$ =0.1 ignition/person*million hectar for a grid cell with the most theoretically wealthy population (WI=30 – closest is the Stockholm metropolitan area). Average value of $\bar{a}$ is equal to 0.22 ignition/person*million hectar (WI=20.5) for peninsular Spain as in Reg-FIRM (Venevsky et al., 2002).





Total number of human fires in a grid cell is calculated as Eq. (10):

$$N_{fires_{hum}}(d) = FDI\,(d) * (N_{ignitions\_hum_{rur}} + N_{ignitions_{urb}}) * S_{grid} \qquad (10)$$

The number of human fires for peninsular Spain was validated for Reg-FIRM, which has the same equations as SEVER-FIRE in the region. To check plausibility of approach for calculation of total number

of fires, we make validation for Canada for 1961-1995 (Stocks et al., 2002) (see Fig. 3), because Canada has significant variation for climatic conditions, vegetation composition, poulation density and socio-economic state of population.

The description of human ignitions in SEVER-FIRE model is very simplistic and does not have intention to describe to major extent complex economic, cultural and social practice of people (agricultural, hunting

or pastoral, other) resulting in pyrogenic activities. We left out (or oversimplified, like in the timing function and mathematical expectation of number of ignition produced by one person) description of an influence of land use to number of human ignitions in the fire model, because SEVER-DGVM anyway does not include description of human land use and/or it's influence to natural vegetation. By application of SEVER-DGVM we aim to describe relatively human-less global vegetation, which got additional

control regulator, namely external human and/or lightning ignitions. This limitation of SEVER-DGVM implies certain constraints on our results in both vegetation distribution and areas burnt, but it also gives us an opportunity to identify and locate the areas, where interaction between land use, fire regimes and vegetation should be described explicitly and accurately.

### 4) *Simulation of fire spread after ignition*

Rate of fire spread after an ignition event is simulated using a simplified version of the Rothermel thermodynamic equation (Venevsky et al., 2002), and depends on wind speed, fuel  bulk density and soil moisture content in the upper layer as a proxy of fuel moisture. As in the Reg-FIRM approach, a fire cannot take place when fuel loading threshold (100 g/m$^2$), calculated as litter pool by a DGVM, is not crossed. Simulation of rate spread, using Rothermel equation, in SEVER-FIRE is similar to the one used

by some of recent landscape fire models (Cary et al., 2006) and other global fire models. However, there is a large difference in translation of rate of fire spread into areas burnt in landscape models and SEVER-FIRE. Indeed, landscape models account for terrain and fuel discontinuity (water bodies, highways etc), while global fire models do not include this feature. Analysis, to which extent up-scaling from landscape level to a rather coarse grid cell of SEVER-FIRE should be done in the future.

### 5) *Fire termination*



Fire termination occurs with the onset of a significant rainfall event (more than 3 mm), causing weather danger to drop to zero. Close to cities, fire termination occurs after a delay dependent on distance to the city, as a proxy for human fire suppression. Fire suppression function (time to eliminate a fire) was constructed as a log-linear regression function from distance to the city, using fire duration statistics for

Europe and Russia from EFIC database (San Miguel-Ayanz et al., 2012). As a result, a single fire can continue in the model from one hour up to two days (see Eq. (11)):

$$\overline{duration} = 2 * (1 - \exp(-11 * 10^{-3} * dist),\qquad(11)$$

where *dist* is a distance (km) from a nearest city (area with $P > 400$ persons/km$^2$).

### 6) *Estimation of fire effects*

The rate of spread is converted to an absolute value of average area burnt for one fire, using elliptic fire spread model (van Wagner, 1969) similarly to the Reg-FIRM approach (Venevsky et al., 2002), which is also adopted by majority of other global fire models.

Daily area burnt in the DGVM grid cell is calculated as Eq. (12):

$$aburnt(i) = N_{fire}(i) * S(i) + N_{fire}(i-1) * S(i),\qquad(12)$$

where $N_{fire}(i)$ are number of fires, ignited in a day $i$, $N_{fire}(i-1)$ are number of fires continuing from previous day (if any do exist) and $S(i)$ is an area of spread for one fire, determined by vegetation and climate (see above).

Daily burned area estimates are aggregated annually to estimate fire effects. Percentage of vegetation individuals killed depends on area burned and on resistance of each PFT to fires (Table 1), taken directly

from the Glob-FIRM (Thonicke et al., 2001). The percentages are then converted to emissions, based on vegetation carbon content (dead PFT individuals are considered to be entirely burned), and daily redistributed following the profile of fire probability.

The model outlined above should be considered as a first approach to design a global completely process-oriented fire model based only on field observations and physically based assumptions with no satellite

derived functions. Still more analysis to be done for representation of fire processes within the model and calibration of parameters used in the model. For instance, study of Scott and Burgan, 2005 indicated that moisture of extinction, used in SEVER-FIRE (see Table 1) may vary from 12% to 40%, for different fuel



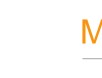 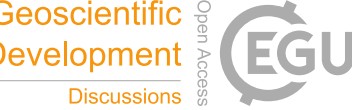

types, i.e. has a larger range than in our model. We plan to make sensitivity and optimisation tests to improve the SEVER-FIRE model parameters and modifications of equations when necessary.

## 2.2 Data

### 2.2.1 Climate data

For this study, precipitation data from the National Centres for Environmental Prediction (NCEP climate data (minimum/maximum temperature, precipitation and convective precipitation, short-wave radiation and wind speed, http://www.cpc.ncep.noaa.gov/) were interpolated to 0.5 degree longitude/latitude spatial resolution for the period 1957 to 2006 (52 years). Daily wind speed is not well estimated in reanalysis approach (Kalnay et al., 1996a), so it was averaged over the entire period and applied in simulation runs

without inter-annual variability. The input soil texture data and $CO_2$ atmospheric concentration over the same period coincide with those of the LPJ-DGVM (Sitch et al., 2003). The model is run globally from bare soil state 15 times with the climate data for 52 (years and the $CO_2$ atmospheric concentration fixed for the year 1957 (spin-up period), in order to achieve equilibrium of soil carbon pools. From this equilibrium state, SEVER is forced by climate and atmospheric $CO_2$ for the period 1957- 2006 (transient

period).

### 2.2.2 Socio-economic data

Distance from a city was pre-calculated from population density and the ratio of urban to rural population. For this, areas where urban population density exceeds 400 persons/km² were considered as cities (UN definition of Human Settlements Program). Gridded population and rural to urban ratio data sets for years

1940-2050, used by SEVER-FIRE were elaborated using UN Development Program estimates by major economic regions. Wealth index was calculated first for 600 cities around the globe (Svirejeva-Hopkins, personal communication) using approach of UN Human Settlement Program as a sum of six socio – economic components, each normalized to be ranged between 0 (minimum) and 5 (maximum). Components included GDP per capita, number of persons with high education, number of doctors, crime

rate, access to clean potable water, air pollution level. Data for the cities was extrapolated for the entire land area using non-linear spline at a regular grid of DGVM.

### 2.3 Burned area and carbon emission validation data

GFED is a global 1º resolution database (van der Werf et al., 2006a), which relies on three different active fire products calibrated to Moderate Resolution Imaging Spectrometer (MODIS) 500 meter burned area,

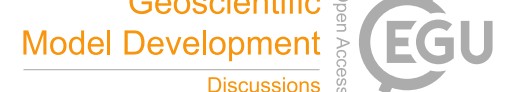

for a temporal coverage spanning 1997-2006 (Giglio et al., 2006). Fire activity data from the Tropical Rainfall Measuring Mission (TRMM) – Visible and Infrared Scanner (VIRS, (Giglio et al., 2003)) and European Remote Sensing Satellites (ERS) Along Track Scanning Radiometer (ATSR, (Arino and Plummer, 2001)) sensors are used for the 1997-2001 period. Over 2001-2006, the calibration was based

on active fires from MODIS (Giglio et al., 2006). Carbon emissions were then estimated based on those burned area estimates, with fuel loads calculated by the Carnegie-Ames-Stanford Approach (CASA) model (van der Werf et al., 2006a).

The active fire to burned area calibration step and the use of three different sensors to build this dataset generate significant uncertainties on burned area estimates, which are considered to be about 50% at

regional scales, although not quantified in the current version of GFED (van der Werf, personal communication). Emission uncertainties are consequently higher, taking into account their further dependence on the CASA model and on fuel loads and emission factor values.

### 2.4 Fire incidence, fire variability and carbon emission evaluation

We chose to focus primarily on burned area to evaluate the model at global scale, as this is a prerequisite

to estimate carbon emissions. However, carbon emission being an essential aspect of biomass burning, its representation is briefly evaluated.

Fire incidence, seasonality, and inter-annual variability from SEVER are compared to GFED data over the 1997-2006 period. As a DGVM, SEVER considers grid cells to be 100% land or water. This required a few adjustments on both datasets (not discussed), causing minor changes in the original GFED statistics.

We consider burned fraction (BF) rather than burned areas, a latitudinal unbiased indicator of fire density given the use of a lat-lon grid.

Fire incidence is mostly dependent on three key factors, conceptualised by the fire triangle (Schoennagel et al., 2004): fuel availability, readiness of fuel to burn, and ignition source. SEVER spatial patterns of fire incidence are first compared to GFED, through the mean annual grid cell burned fraction (BF). BF

drivers are then explored with a selection of relevant environmental variables, based on the fire triangle concept:

-      Annual amount of precipitation, from the CPC merged Analyses of Precipitation (CMAP, (Xie and Arkin, 1997)), provided by the NOAA/OAR/ESRL PSD, Boulder, Colorado, USA, (http://www.cdc.noaa.gov/).





- An indicator of dry season severity (DSS), which was constructed from precipitation (CMAP) and temperature data (NCEP/NCAR re-analysis project, (Kalnay et al., 1996b)). The indicator (Breckle, 2002), representing a rainfall deficit or a temperature excess, is computed as indicated by Fig. 4. Here we consider it as a rainfall deficit (unit: mm).

5 - Net Primary Productivity (NPP). Its influence on fires is estimated with NPP estimates from (Imhoff et al., 2004) and from SEVER.

- Land cover spatial distribution. SEVER-DGVM vegetation distribution and its impacts on BF patterns is evaluated with the Global Land Cover for the year 2000 (GLC2000, (Bartholomé and Belward, 2005)).

10 - Human rural and urban population density from the Global Demographic Data Collection (Vorosmarty et al., 2000), provided by the University of New Hampshire, EOS-WEBSTER Earth Science Information Partner (ESIP). An indicator of the rural predominance of the population was defined (Eq. (13)):

$$Rurality = \frac{rpop-upop}{rpop+upop},$$ (13)

15 where *rpop* and *upop* being respectively the rural and urban population of the considered grid cell. *Rurality* varies between 1, fully rural, to -1, fully urban populations.

- Gross Domestic Product (GDP) gridded data (van Vuuren et al., 2007), provided by the Netherlands Environmental Assessment Agency.

We left aside wind speed, which significantly affects readiness of fuel to burn and fire spread, as an 20 analysed environmental variable, due to constrains put on it in the presented SEVER-FIRE simulations (see description of input data in Sect. 2.2).

We used CMAP precipitation data (extracted mainly from remote sensing data) in analysis to get more realistic relationship between fire regimes observed and precipitation. We, however, could not use CMAP precipitation as climate input for SEVER-DGVM due to too short period of observations (CMAP started 25 from 1979) and used instead the NCEP reanalysis precipitation data which is longer and provide bigger ratio between lengths o transient and spin-up simulation periods in DGVM important for realistic description of vegetation and fires. Thus, discrepancies in relationships between fire and precipitations in our analysis for GFED and SEVER-FIRE cases can be to some extent explained by differences between



NCEP and CMAP precipitation fields. These differences, however, have only regional character and do not change our general conclusion.

The relationship of chosen variables with fire incidence is not linear, and it involves multi-variable interactions. A more in-depth analysis of fire drivers would thus benefit from the use of multivariate statistics. We chose to avoid this level of complexity, since the most important conclusions are likely to be drawn from straightforward analysis, as a first evaluation of a global fire model. We thus analyse fire incidence through simple bi-dimensional plots.

Seasonality is evaluated via the fire season peak, i.e. the month with maximum fire activity for each grid cell. Inter-annual variability is compared to GFED both globally and regionally, to identify how the model performs on specific fire events and for different ecosystems. Again, in a similar way to fire incidence, fire inter-annual variability has been shown to depend on climatic and vegetation conditions. (Meyn et al., 2007) highlight three types of fire ecosystems, depending on their annual fire limitation by fuel amount, readiness of fuel to burn, or both, considering that the availability of ignition sources is relatively constant in time. Here, we further explore the climate impact on the readiness of fuel to burn, analysing the implications of both fire season precipitation and fire season maximum temperature for fire inter-annual variability, along three ecosystem types (boreal, tropical humid, and dry/semi-dry). To extract those variables, the extent of the fire season in a grid cell was defined as the months with more than $1/12^{th}$ of the mean annual BF. Fuel availability, the second factor highlighted by (Meyn et al., 2007), is also discussed.

## 3. Results

### 3.1 Fire incidence and emissions

Figure 5 shows the spatial distribution of the average annual BF for GFED and SEVER. GFED clearly depicts the most extensively burned continents, i.e. Africa and Australia. It also indicates high fire activity at the edges of the tropical forest, due to land clearing and pasture management, in Central and South America and South East Asia (Langner et al., 2007; Morton et al., 2006). Fire incidence is much lower in most temperate and boreal ecosystems, except for the north-western Iberian Peninsula and Kazakhstan, both regularly affected by fires. A few other regions display high BF values, for example eastern Siberia and Alaska. Note, however, that for ecosystems with a long fire return interval, as is the case in boreal regions, the statistics computed over 10 years are very sensitive to the occurrence of important fire events during that period, and it can not be considered representative of the long term regional fire regime.





Eastern Siberia, for example, was highly affected by fires in 1998, boosting the 10 years average (Kajii et al., 2002; Le Page et al., 2008).

SEVER accurately reproduces some of the main spatial patterns of fire incidence, i.e. high BF values over Africa and Australia, very limited fire activity in the tropical evergreen forest and in most temperate and
boreal regions. For a better emphasis of the discrepancies, Figure 6 illustrates the mismatch between GFED and SEVER through a normalised difference burned fraction index (NDBF) computed as Eq. (14):

$$NDBF = \frac{BF_{SEVER} - BF_{GFED}}{BF_{SEVER} + BF_{GFED}},$$    (14)

Where $BF_{SEVER}$ and $BF_{GFED}$ are the annual fire incidence averaged over 1997-2006 from the model and the observations, respectively. $NDBF$ is constrained between -1 (large model under-estimation) and 1
(large model over-estimation). Finally, Figure 7 shows the gradient of three broad PFTs classes (Bare soil, Grass and Trees), as modelled by SEVER, and the regions of large over/under estimation of the actual tree cover percentage inferred from GLC2000. Those results and further comparison with GLC2000 clearly reveal the following patterns:

-    Regions with low observed fire incidence and the presence of grass in the model display fire over-
15        estimation, regardless of the GLC2000 landcover, and the more grass, the higher the over-estimation. This is the case for example in North America, India, South America and Papua New Guinea.

-    Regions with dominant tree cover, or with a large over-estimation of trees in the model, display under-estimation of fire incidence. This is the case in a large strip covering Kazakhstan and eastern Europe, and in most of South East Asia, for example.

-    The model underestimates the very high fire incidence observed in sub-Saharan Africa.

Considering drivers of BF spatial distribution, Figure 8 illustrates the interactive influence of paired combinations of the previously described variables. In GFED, the most affected regions are clearly constrained by annual precipitation between 500 and 1500 mm/year and a dry season severity ranging from 150 to 500mm of rainfall deficit (Fig. 8: top). SEVER is less restrictive regarding this climatic
limitation, but the general dependence patterns are similar to the observations. Concerning vegetation characteristics (Fig. 8: middle), fires affect ecosystems of all levels of NPP, although fire incidence is low at the extreme ends of the spectrum. Similar values of NPP and annual precipitation can be found in very different ecosystems, as in boreal and sub-tropical regions for example, with great differences in fire incidence, hence the low predictability of GFED BF by NPP and precipitation. SEVER also shows little





constraining of the mean BF by the combination of those two variables. Finally, high fire incidence is biased towards rural regions with very low economic income (<600 US$/capita/year), as shown in Fig. 8: bottom, with the exception of Australia, the only wealthy country highly affected by fires. SEVER also shows this rural bias, but on average allows higher fire incidence in wealthy regions, including North

America.

Finally, Figure 9 displays the mean annual carbon emissions for GFED and SEVER. Emissions are mainly dependent on fire incidence, the type and moisture content of the affected vegetation, and fire severity. In SEVER, dead PFTs individuals are entirely emitted to the atmosphere, while GFED takes into consideration combustion completeness. Consequently, the absolute level of emissions cannot be

compared, being much higher in SEVER, as expected. However, the spatial patterns reveal the importance of tropical savannas and forests in the global partitioning of carbon emissions in both GFED and SEVER, as well as a significant contribution from boreal regions. We are planning to correct SEVER for combustion completeness as well as for post-fire mortality processes.

### 3.2 Seasonality

Figure 10 shows the spatial patterns of the month with maximum fire activity for each grid cell, and the mismatch between GFED and SEVER. SEVER roughly reproduces the observed spatial patterns, with 73% of the grid cells with a mismatch lower than or equal to 2 months. Significant discrepancies occur in Sub-Saharan Africa, which peaks over March to June in the model, while GFED, along with other observation sources, indicate October to February (Barbosa et al., 1999; Clerici et al., 2004; Dwyer et al.,

2000b).

Sub-Saharan Africa is a major fire region (Dwyer et al., 2000c; Tansey et al., 2004), contributing to a large fraction of global fire activity from October to February, a period when most other regions experience little or no fire activity. As such, the inability of SEVER to reproduce fire seasonality in Sub-Saharan Africa is one of its major current limitations. Delayed fire season is also significant in Central

North America and south-eastern Australia.

The fire seasonal cycle is partially driven by climate, but it can also be strongly influenced by human activities. Figure 11 illustrates the averaged profile of the fire season and the dry season over Sub-Saharan Africa, for those grid cells with a SEVER fire peak discrepancy larger than or equal to 4 months. For each of these cells, we computed the monthly fire season, centred the peak month on the x-axis, and then

derived the corresponding monthly DSS profile. Once averaged over all grid cells, the fire and DSS





profiles show the temporal connection between both variables. Figure 11 clearly indicates that in the grid cells considered, the fire season is shifted towards the early dry season in GFED, and towards the late dry season in SEVER.

In regions with lower use of fire as a management tool, as in boreal forests, the model performs much
better and, along with the observations, tends to place the peak month in the middle or late dry season (not shown). The implication of these findings for model improvement are detailed in the discussion section.

### 3.3 Inter-annual variability

Figure 12 shows the grid cells correlation between annual BF timeseries from GFED and SEVER.
Equatorial Asia, Mexico and a majority of boreal regions are in good agreement, along with part of South America. As discussed later, those regions are characterised by their sensitivity to climate variability, especially to the El Niño of 1997/98 (Le Page et al., 2008). Poorest agreement is found in Africa, India, China, western Russia, south of the USA Great Lakes, and in parts of South America.

Inter-annual variability is further analysed using a set of 13 regions, as represented in Fig. 13. Globally,
and for each of those regions, Figure 14 shows the BF inter-annual anomalies from GFED and SEVER, along with the monthly distribution of fire activity as a further indicator of the timing of specific fire events, and of fire seasonality. The very poor agreement in the global plot was to be expected, given the discrepancies in mean spatial fire incidence (Fig. 5), resulting in different contributions from regions to the total fire anomalies. This is clearly revealed by the monthly plot, showing that total fire activity in
December-February, peaking in GFED with the large contribution of sub-Saharan Africa, is very low in SEVER. Consequently, a given fire anomaly in Africa has a much bigger global impact in GFED than in SEVER.

Regional partitioning allows identifying and comparing specific fire events more easily, especially the ones driven by large scale climatic variability. The El Niño episode of 1997-1998 appears clearly in the
BONA, CEAM, BOAS and EQAS regions in the observations, and is generally captured by the model with precise timing. Annually, the importance of those events is also reproduced for EQAS and BOAS, with respectively 1997 and 1998 being the peaking year in both GFED and SEVER. Generally, fire patterns in the other regions are not properly represented. The monthly resolution plots also give further insights into the regional scale seasonal cycle, which is generally very well reproduced, except for
northern hemisphere Africa and Australia.





Figure 15 displays the dependence of fire anomalies on precipitation and temperature anomalies over the fire season, through their effect on soil and vegetation moisture status. Drought conditions are the main pre-requisite for fire occurrence within all vegetation types, although in low NPP ecosystems, low vegetation amount can be a limiting factor, resulting in a dependence of fire anomalies on growing season

precipitation also (Holmgren et al., 2006; van der Werf et al., 2008). The relationship is first pictured globally (Fig. 15), showing that both precipitation and temperature anomalies are strong drivers, constraining positive fire anomalies almost exclusively to precipitation deficits, and towards positive temperature anomalies. This relationship is then analysed in GFED for 3 types of ecosystems:

-    Boreal ecosystems, a spatial aggregation of the BONA and BOAS regions. Boreal fires are shown
10        to be strongly dependent on temperature, at a level comparable to precipitation.

-    Tropical humid regions, selected within South America, Africa and Equatorial Asia, as the pixels with annual precipitation above 1500mm. Their fire anomalies are also strongly related to precipitation, while temperature is a weak driver.

-    Semi-dry and dry African and Australian regions (annual precipitation below 500mm), which are
15        characterised by high anthropogenic fire activity. For those regions, both fire season precipitation and temperature anomalies are poor predictors of fire anomalies.

Those patterns are well reproduced on a global scale, such that the patterns of dependence on both climatic variables are similar in the model and in observations (Fig. 15). In boreal/tropical humid ecosystems, SEVER shows the same trends towards more/less dependence on temperature, although not as neatly as

in GFED. In the case of semi-dry and dry African and Australian regions, the model also shows a weaker dependence on precipitation and temperature, but stronger than in the observations.

### 4. Discussion

Perhaps one of the most important achievements of SEVER, as revealed by this study, is the realistic modelling of strong climate driven fire anomalies, such as the large biomass burning events resulting

from El Niño-induced droughts in various regions of the world (Fig. 12 and Fig. 14). This climate induced variability is known to be considerable and has important consequences for atmospheric composition, the terrestrial carbon cycle, and biodiversity, as discussed in the Introduction. As such its accurate representation in DGVMs and ESMs is essential.

The in-depth analyses of this climatic influence highlight the variability of the precipitation/temperature
dependence patterns (Fig. 15). Boreal regions are characterised by great annual amplitudes of



precipitation and temperature. As such, both play an important role in the dynamics of soil and vegetation moisture status, through rainfall and evaporation, thus the strong fire dependence on both variables. In tropical humid regions, temperature variability is much lower, and only a major and prolonged precipitation deficit will result in fire prone conditions (van der Werf et al., 2008).

Finally, semi-dry and dry regions of Africa and Australia are characterised by a low dependence on both parameters. Those regions are under specific climatic conditions, characterised by a rather short and irregular wet season for vegetation growth, followed by a long dry season (Peel et al., 2007). Under those conditions, fuel availability, rather than its readiness to burn, limits the occurrence of fires (Meyn et al., 2007). Under low wet season precipitation, vegetation build-up may be too low to sustain a fire. Under

high wet season precipitation, vegetation growth leads to less patchy vegetation, which will dry out over the following dry season, becoming highly susceptible to fires. This scheme is very specific of those hot dry and semi-dry regions dominated by annual herbaceous vegetation. In the case of middle to high productivity ecosystems with the presence of woody vegetation, the relationship is generally reversed: enhanced wet season precipitation leads to higher soil and vegetation moisture status, delaying desiccation

over the dry season, thus reducing fire susceptibility. The contrast between those two distinct vegetation-climate-fire relationships is most evident in Australia (Fig. 16). The SEVER vegetation scheme did not perform very well over Australia, and so the role of wet season precipitation is not properly represented (not shown).

At global scale, SEVER is shown to be fairly realistic regarding this temperature/precipitation dependence,
which was to be expected since both variables are involved in the fire weather danger and fire spread calculations. However, the variability of the relationship along ecosystem types (boreal, tropical humid, semi-dry/dry), resulting from complex interactions between fire drivers, is not as straightforward to capture. The realistic results for such an interactive system suggests that the feedback mechanisms as defined in the SEVER-DGVM/SEVER-FIRE coupled scheme do reach a reasonable level of complexity
and accuracy, especially in the case of boreal and tropical ecosystems.

The mean burned fraction (Fig. 5) is a more challenging feature for the model to replicate. Key associations represented in the fire triangle (Schoennagel et al., 2004) are, however, reproduced (Fig. 8), i.e. fire occurrence limitation by moisture in very humid ecosystems, or by low fuel amount in arid regions. Unfortunately, SEVER models potential - not actual - vegetation cover, hampering an in-depth diagnostic
of the fire incidence estimates. However, grass/trees appear to be over/under sensitive to fires, with the exception of highest fire incidence regions (Africa, northern Australia), where SEVER underestimates fire activity, independent from vegetation cover (Fig. 6 and Fig. 7). The main PFT parameters controlling





fire incidence are bulk density (fire ignition and spread, see Table 1), and flammability (fire danger index computation). Flammability takes the same value for all tree PFTs, and a distinct value for both C3 and C4 grasses together. As such, it may be a relevant factor to correct the over/under estimation observed in grass/trees. Of critical importance for fires are also three vegetation types not yet included in SEVER-
DGVM: croplands and pasture (land management fires (Pyne, 2001)), savannas, and peatlands (modest land extent, but major carbon hotspot (Page et al., 2002; Turquety et al., 2007)).

It is also essential to improve our understanding of anthropogenic impacts on fire incidence. The initial assumptions of the model, with population and wealth status as the most important human proxies, are to be re-assessed carefully in regional studies, given the implication of other factors. Especially, the most
evident cases of human induced increased or decreased fire activity are related to land use type and agricultural practices, more than to economic and social status. In Africa for example, the combination of a strong seasonal wet-dry climate with regular human ignitions favours high fire incidence. Thus, a simple timing function for rural population implemented into SEVER-DGVM may not work properly in Africa. Relating those ignitions to low wealth status, as done in SEVER, is certainly functional after a few
adjustments, but seems less robust to other regions than an association of land use with timing of human pyrogenic activities and number of human ignitions. As an illustration, wealth status is not well adapted to account for high fire incidence induced by humans in northern Australia (Russell-Smith et al., 2007). Additional proxies for human pyrogenic activities implemented in SEVER-FIRE could include deforestation activities (Zhan et al., 2002) and land use/landcover data (Thenkabail et al., 2006).

Advantages of including relationship between land use and timing of pyrogenic activities in SEVER would possibly also extend to a better representation of fire seasonality. In sub-Saharan Africa for example, Figure 11 reveals that the fire season (October-February, Fig. 10) is shifted towards early months of the dry season, which mainly results from the use of fires for agricultural and land management practices (Clerici et al., 2004). For the whole southern hemisphere, however, human pyrogenic activity
in SEVER is set to reach a maximum from March to May and September to November, which is not realistic in the case of sub-Saharan Africa, a major fire region. Timing of pyrogenic activities in sub-Saharan may be rather challenging as even implementation of land use in global fire model (Le Page et al., 2015) still brings 1 to 3 month delay in fire peak. Besides, it was demonstrated that religious affiliation modulates agricultural burning activities in the area (Pereira et al., 2015), which is completely off the
view of global fire modelers at the time. It is seen that a set of regional case studies with an active use of available historical data is necessary to implement more realistic features of human pyrogenic activities in global fire models.





Description of lightning fires need also improvements, starting from estimation of number of lightning strikes effective for fire ignition. Despite lightning strike is considered to be to major extent a stochastic event, there is a visible room of better description of number of cloud-to-ground flashes based on recent findings of role of aerosols in electrification of thunder clouds (Stolz Douglas et al., 2015; Venevsky, 5   2014).

## 5. Conclusions

This paper analyses results from a DGVM which includes an interactive, dynamically-linked fire module. It reveals that the most important climate driven fire features are reproduced by the model, while the dependence on vegetation characteristics and, especially, human pyrogenic activities prevents the further
10   development of realistic estimates of fire incidence, and of regional to global inter-annual variability. Regional adjustments of global fire models based on analysis of both historical fire statistics/records and recent satellite observations are necessary for further understanding of global fire dynamics in past, present and future.

## 6 Code availability

15   The model code could be accessed by contacting Sergey Venevsky or Chao Wu.



**Acknowledgements**

We thank Guido van der Werf for providing the GFED data and for helpful comments on the manuscript. This study is funded by the Marie Curie Research Training Network GREENCYCLES, contract number MRTN-CT-2004-512464 (www.greencycles.org), National Science Foundation of China (31570475), and Tsinghua University-Saint Petersburg Imperial Polytechnic University Joint Scientific Research Fund (20173080026).

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



**Table 1: 5 of the 35 parameters defined for each of the 10 SEVER PFTs.**

| PFTs | Moisture of extinction[1] | Fire resistance index[2] | Minimum coldest monthly mean T°C[3] | Maximum coldest monthly mean temperature[3] | Bulk density of fuel kg/m2 |
|---|---|---|---|---|---|
| Tropical Broadleaved evergreen tree | 0.3 | 0.12 | 15.5 | ∅ | 3 |
| Tropical Broadleaved rain green tree | 0.3 | 0.5 | 15.5 | ∅ | 2 |
| Temperate Needleleaved evergreen tree | 0.3 | 0.12 | -2 | 22 | 10 |
| Temperate Broadleaved evergreen tree | 0.3 | 0.12 | 3 | 18.8 | 10 |
| Temperate Broadleaved summer green tree | 0.3 | 0.12 | -17 | 15.5 | 10 |
| Boreal Needleleaved evergreen tree | 0.3 | 0.12 | -32.5 | -2 | 16 |
| Boreal Needleleaved summer green tree | 0.3 | 0.12 | ∅ | -2 | 16 |
| Boreal Broadleaved summer green tree | 0.3 | 0.12 | ∅ | -2 | 16 |
| C3 perennial grass | 0.2 | 1 | ∅ | 15.5 | 2 |
| C4 perennial grass | 0.2 | 1 | 15.5 | ∅ | 2 |

[1] *Involved in the computation of fire probability*

[2] *Involved in the computation of vegetation disturbance after a fire*

[3] *∅ indicates no limitation from the considered parameter*



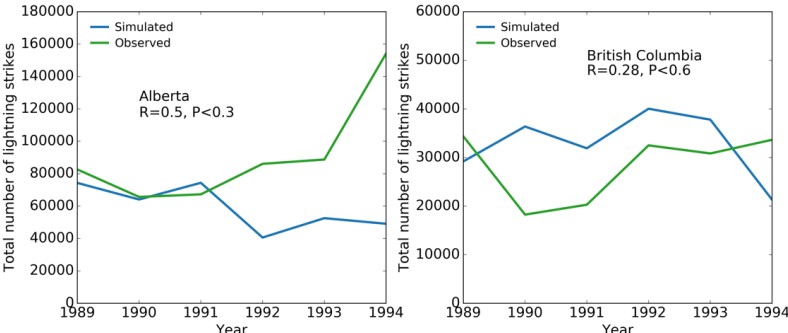

**Figure 1. Observed and simulated number of lightning strikes in central cordillera of Canada. Left: in Alberta; Right: in British Columbia.**

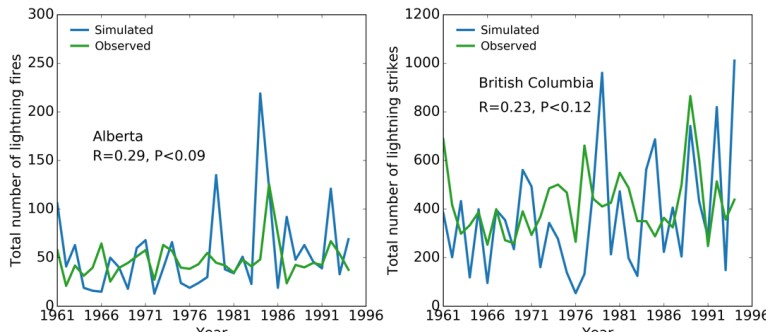

**Figure 2. Total number of lightning fires observed and simulated. Left: in Alberta; Right: in British Columbia.**



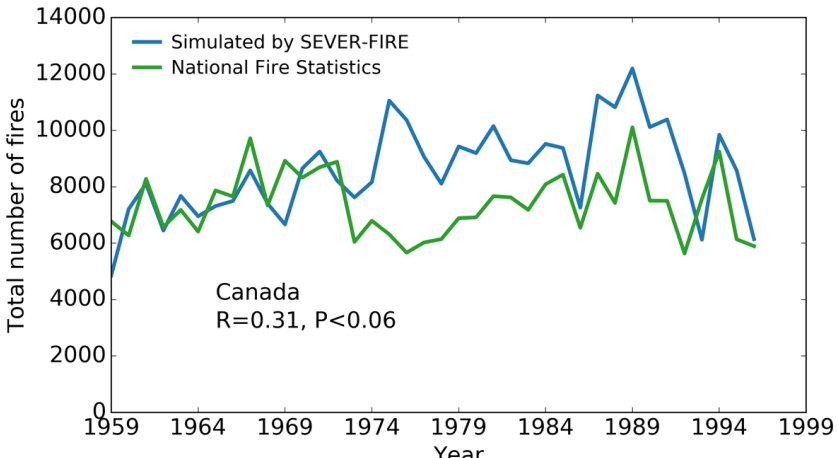

**Figure 3. Registered and simulated number of fires in Canada.**

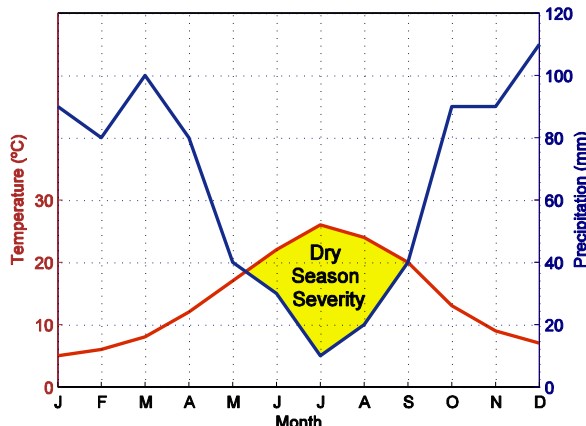

5    **Figure 4: Definition of the dry season indicator on a climatic diagram as the yellow patch area. On the y-scales, 1ºC is equivalent to 2mm/year of precipitation, and Dry Season Severity (DSS) is computed as the area of the region where the temperature profile is above the precipitation profile.**





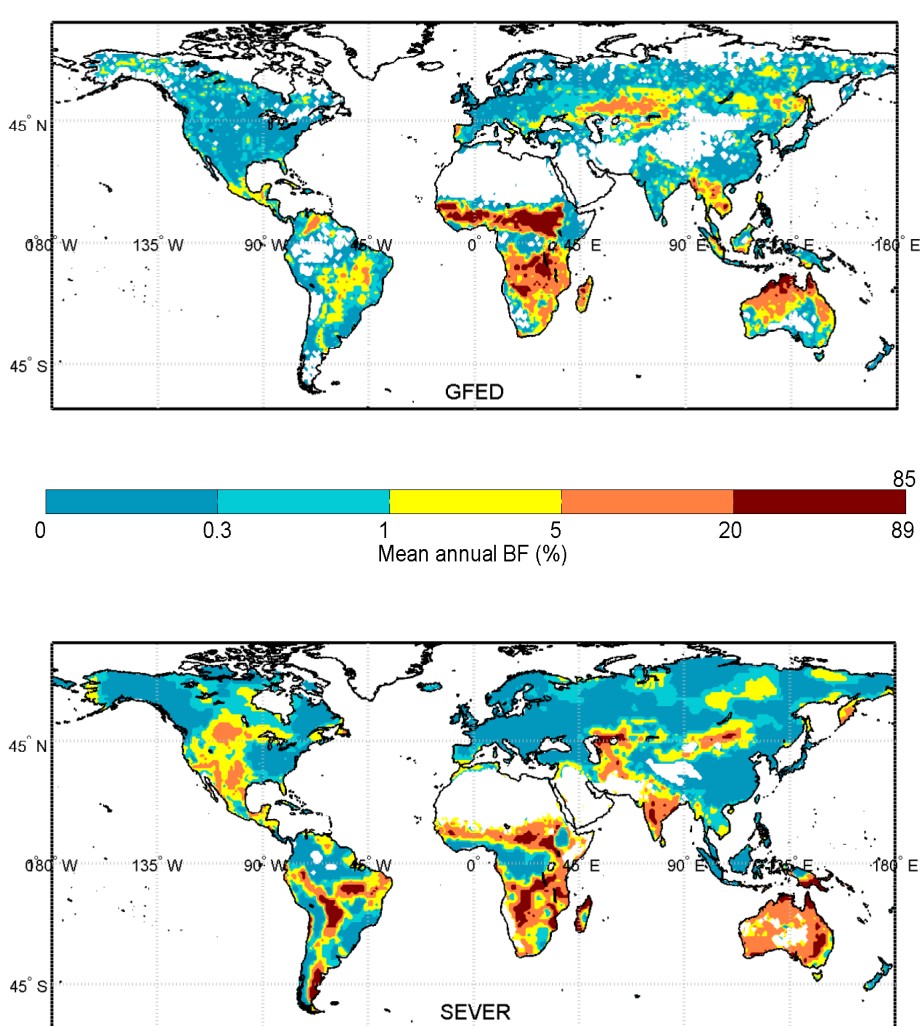

**Figure 5. Mean Annual Burned Fraction (percentage) over 1997-2006. Top: GFED; Bottom, SEVER-FIRE.**

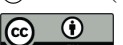



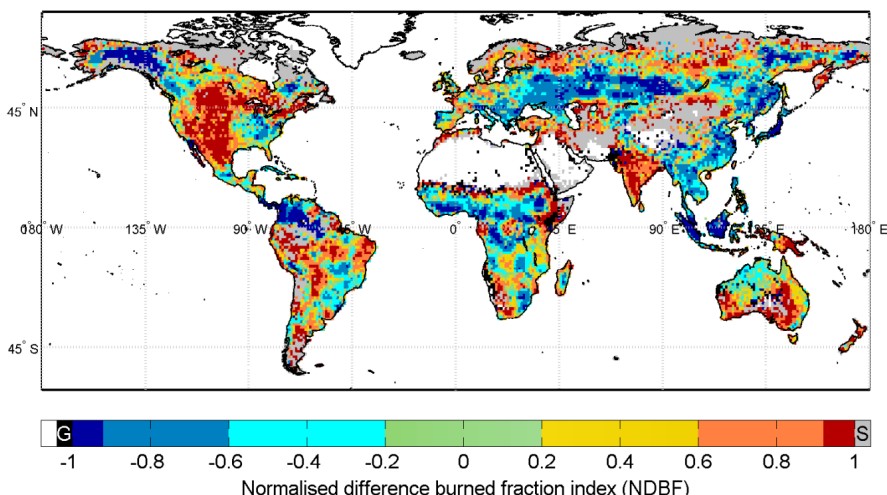

**Figure 6. Discrepancies in the model outputs relative to GFED observation derived data, as represented by the normalised difference burned fraction index (see text). Black/grey colours represent grid cells where fires only occur in GFED/SEVER.**

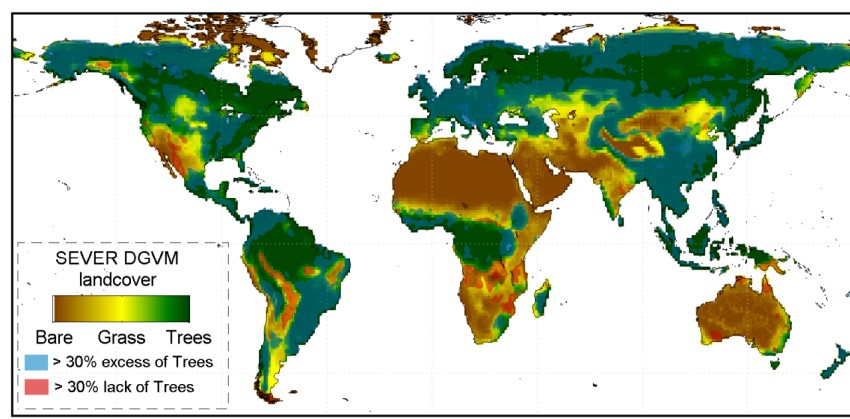

**Figure 7. SEVER DGVM Land Cover distribution, grouped in 3 broad classes: Bare soil, Grass (C3 and C4) and Trees (all Tree PFTs, see Table 1).**





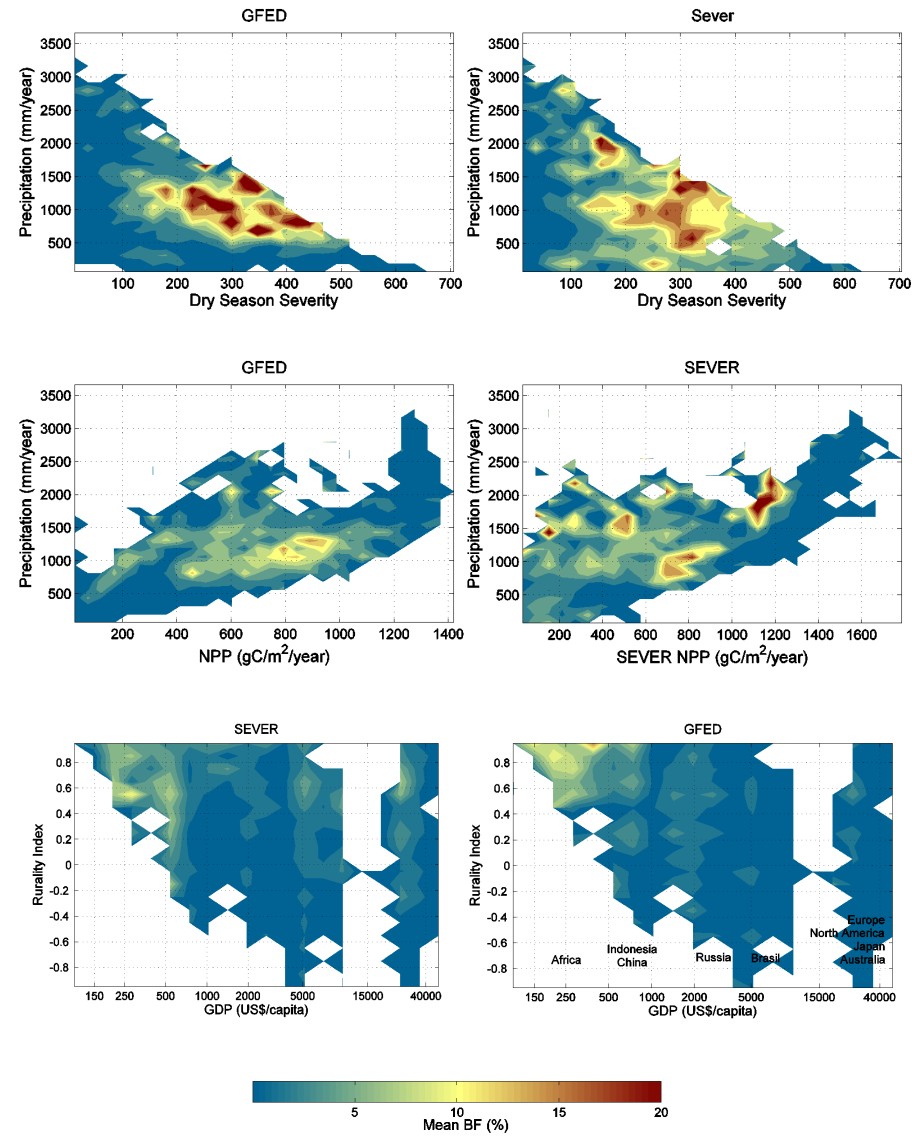

**Figure 8. Mean Annual Burned Fraction over 1997-2006 (left: GFED; right: SEVER-FIRE) as a function of paired parameters. Top: Annual Precipitation and Dry season severity; Middle: Precipitation and NPP; Bottom: Rurality indicator and GDP.**




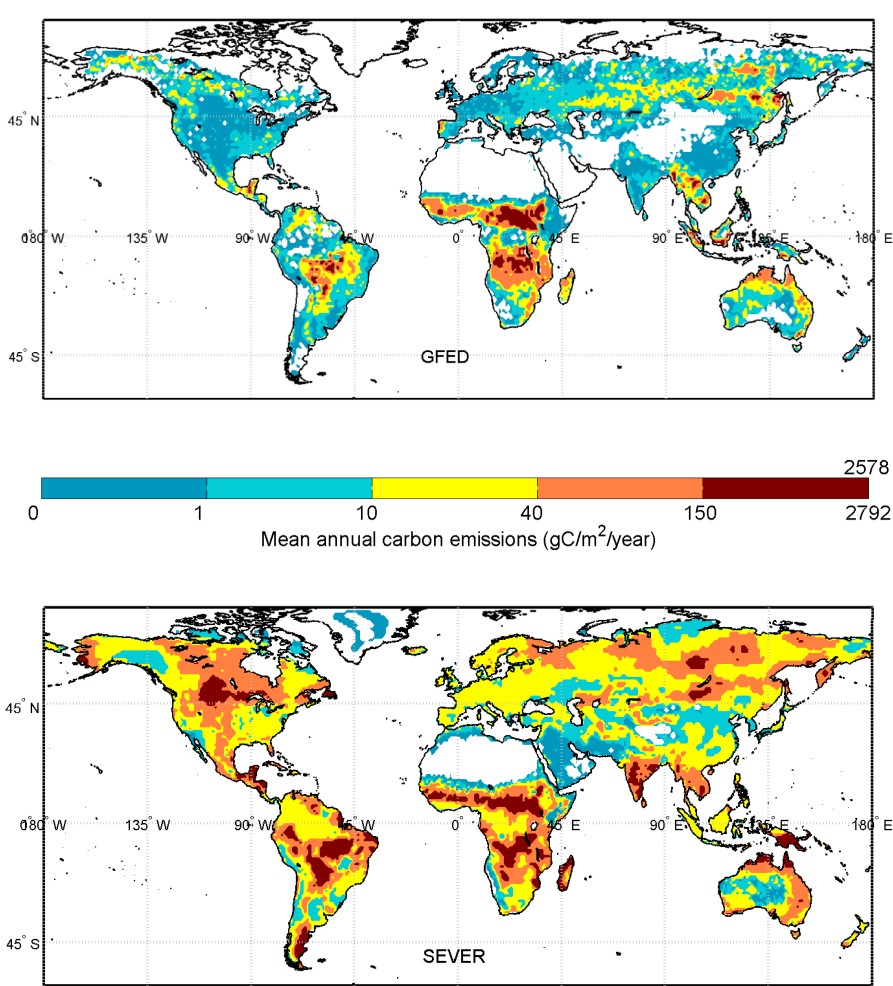

**Figure 9. Mean Annual emissions (gC/m²/year) over 1997-2006. Top: GFED; Bottom, SEVER-**
**FIRE.**



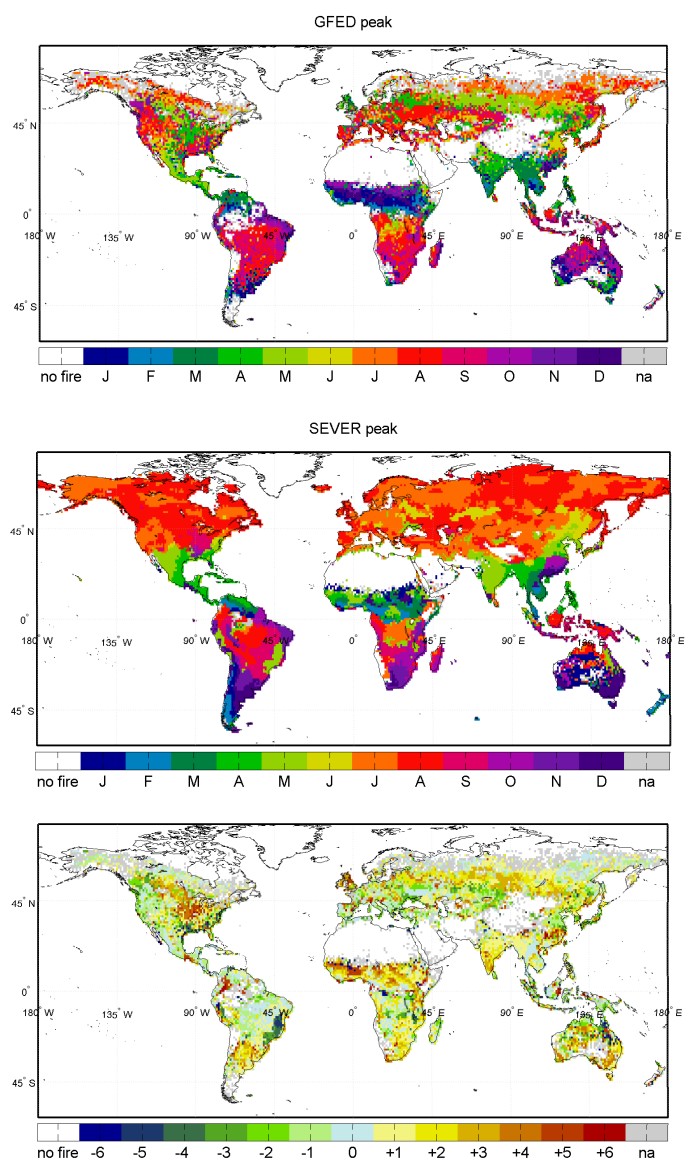

5    **Figure 10. Top: Peak of the fire season in GFED; Middle: Peak of the fire season in SEVER; Bottom: relative mismatch between SEVER and GFED peaking month of the fire season.**





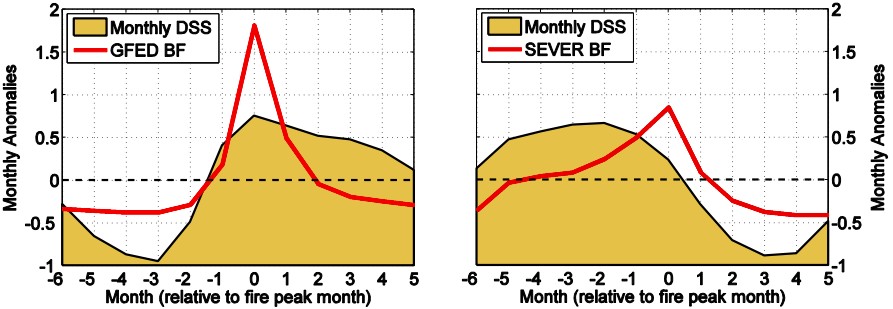

**Figure 11. Averaged correspondence of fire season with dry season anomalies over regions of sub-Saharan Africa with a delay in peak month superior or equal to 4.**



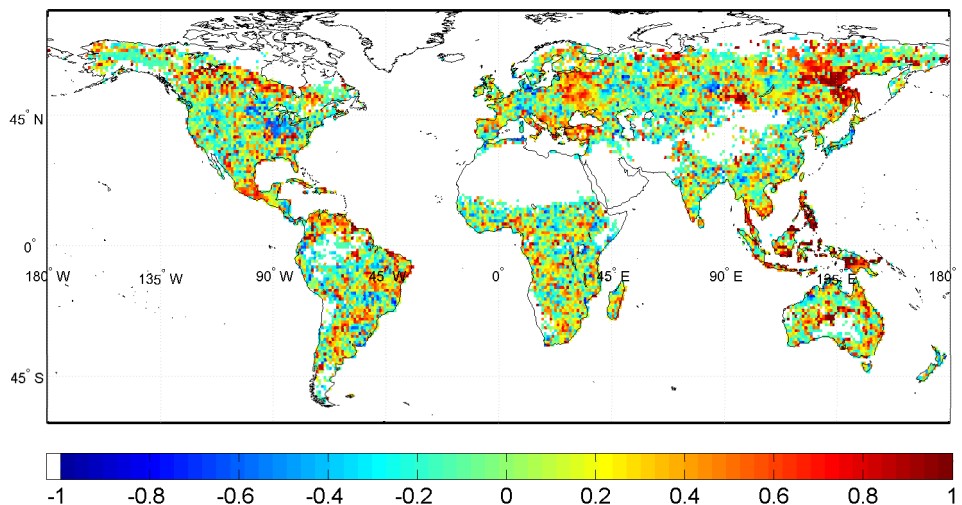

**Figure 12. Correlation of annual BF from GFED and SEVER, over 1997-2006.**

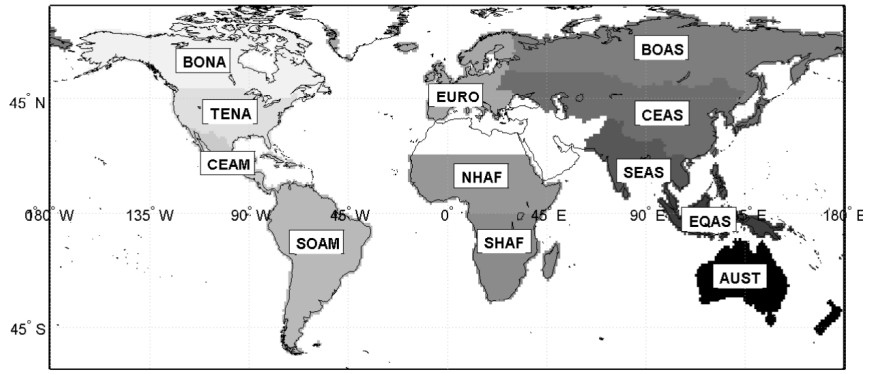

**Figure 13. Regions used for inter-annual variability analysis. BONA: Boreal North America;**
5    **TENA: Temperate North America ; CEAM: Central America ; SOAM: South America ; EURO:**
**Europe ; NHAF: Northern Hemisphere Africa ; SHAF: Southern Hemisphere Africa ; BOAS:**
**Boreal Asia ; CEAS: Central Asia ; SEAS: South East Asia ; EQAS: Equatorial Asia ; AUST:**
**Australia.**







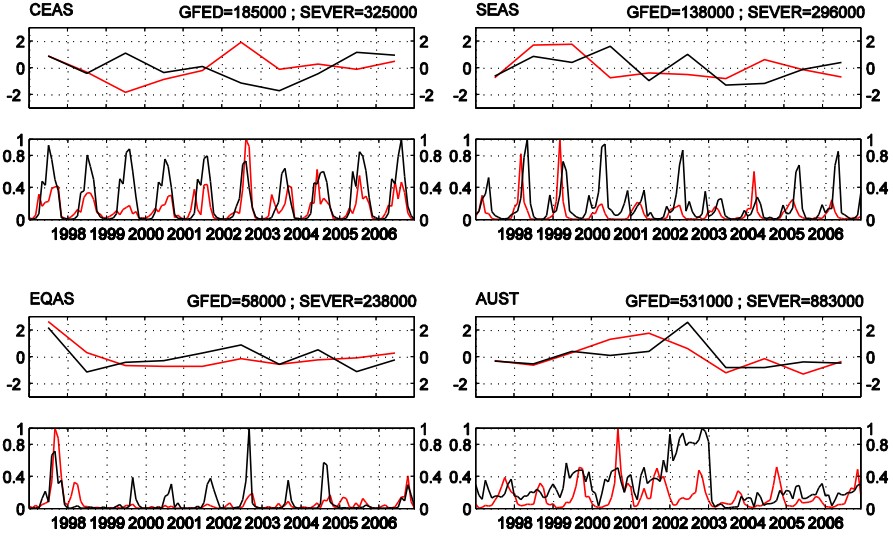

**Figure 14. Regional comparison of fire variability over 1997-2006. For each region subplot: Top: annual Anomalies; Bottom: monthly time series constrained to [0 1]. The region name is indicated at the top left corner, the average fire incidence at the top right.**



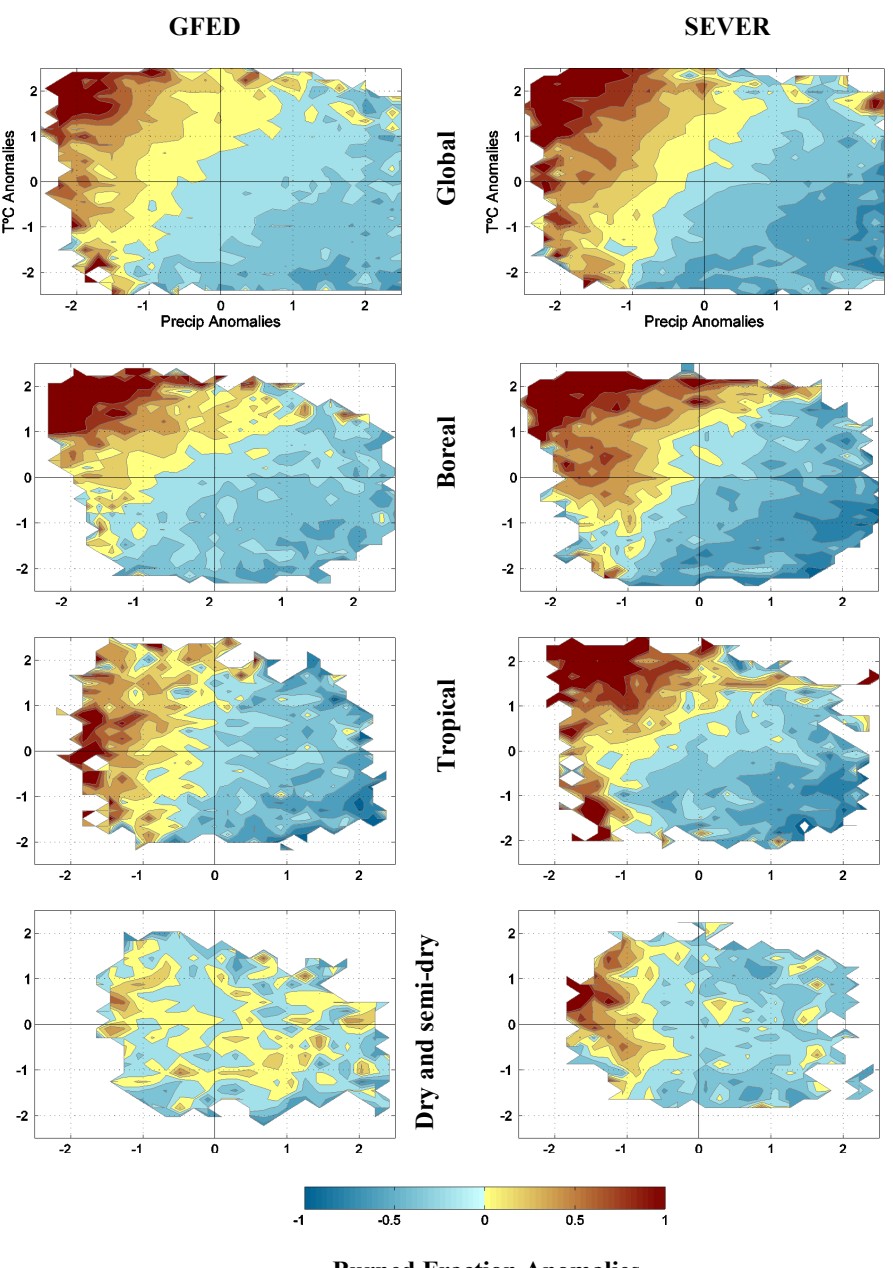

**Figure 15: Dependence of fire anomalies to Temperature and Precipitation**





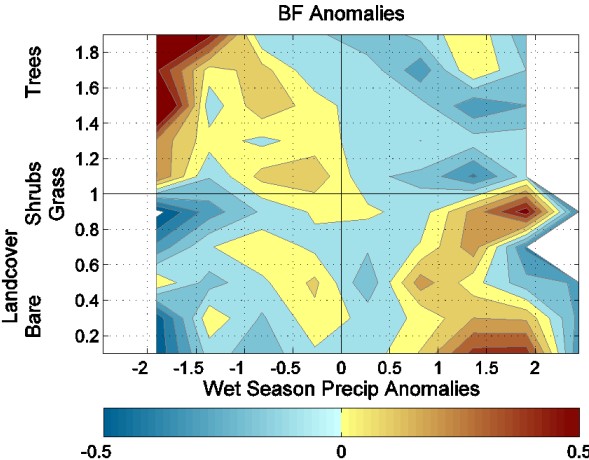

**Figure 16: Dependence of fire anomalies to wet season precipitation and landcover type in Australia for GDED data**