# Peer review of "Analysis fire patterns and drivers with a global SEVER-FIRE v1.0 model incorporated into Dynamic Global Vegetation Model and satellite and on-ground observations."

_Geoscientific Model Development, 2018_

## Short Comment (SC1) · 15 Aug 2018

Dear authors,

In my role as Executive editor of GMD, I would like to bring to your attention our Editorial version 1.1:

http://www.geosci-model-dev.net/8/3487/2015/gmd-8-3487-2015.html

This highlights some requirements of papers published in GMD, which is also available

on the GMD website in the 'Manuscript Types' section:

http://www.geoscientific-model-development.net/submission/manuscript_types.html

In particular, please note that for your paper, the following requirements have not been met in the Discussions paper:

- "The main paper must give the model name and version number (or other unique identifier) in the title."

- "All papers must include a section, at the end of the paper, entitled 'Code availability'. Here, either instructions for obtaining the code, or the reasons why the code is not available should be clearly stated. It is preferred for the code to be uploaded as a supplement or to be made available at a data repository with an associated DOI (digital object identifier) for the exact model version described in the paper. Alternatively, for established models, there may be an existing means of accessing the code through a particular system. In this case, there must exist a means of permanently accessing the precise model version described in the paper. In some cases, authors may prefer to put models on their own website, or to act as a point of contact for obtaining the code. Given the impermanence of websites and email addresses, this is not encouraged, and authors should consider improving the availability with a more permanent arrangement. After the paper is accepted the model archive should be updated to include a link to the GMD paper."

Therefore please provide a reason why your model is not publicly available. If there are no license issues preventing the publication of the code, make the exact version, your article refers to, available via a permanent archive providing a DOI (e.g. Zenodo).

Additionally, add the version numbers of SEVER-FIRE and DGVM to the title of your article upon submission of the revised manuscript.

Yours,

Astrid Kerkweg
* * *

---

## Short Comment (SC2) · 23 Sep 2018

Venevsky et al. (2018) present a new coupled vegetation-fire model, SEVER-FIRE. The presentation of this model is marred by unfounded claims, inconsistencies, and unwarranted disparagement of earlier research.

Readers who closely follow the development of dynamic global vegetation models (DGVMs) will have recognized that the coupled model's parent DGVM is a minor variant of the widely used LPJ model; while the new fire component is based on Venevsky

et al.'s earlier (2002) regional fire model, RegFirm. Thus, the new model is assembled from the same components as the LPJ-SPITFIRE model published by Thonicke et al. (2010), eight years previously. The preprint does not cite more recent developments and applications based on SPITFIRE, such as Prentice et al. (2011), Pfeiffer et al. (2013), Lasslop et al. (2014), Kelley et al. (2014), Yue et al. (2014), Baudena et al. (2015) and Wu et al. (2015).

It is natural to ask what the model achieves that other fire-vegetation models (see e.g. Rabin et al. 2017) do not. The preprint does not claim any superior ability to reproduce observed patterns. It does however claim to be based on a different, and implicitly superior, approach to modelling fire. For example, page 3 refers to other models (including Thonicke et al., 2010) being based on "sets of rather complicated equations with variety of coefficients (despite they name themselves intermediate complexity models)..." Later, the preprint states about SEVER-FIRE that "No satellite derived data are used as an input of the model. Only physically based or just 'common sense' based equations from on-ground observations allow direct implementation of SEVER-FIRE model..." and "Unlike in other global DGVM fire modules ... all equations are kept simple following ideology of Reg-FIRM." Thus, earlier models are criticized for their use of satellite data as input (for lightning frequency in the case of SPITFIRE) although the basis for this objection is not stated. Appeals are made to simplicity and 'common sense' – the former being a defensible aim, the latter not a scientific concept – and, curiously, to an 'ideology'. These appeals do not amount to a convincing case for a new model, especially as it includes equations – notably those determining rates of human ignition, e.g. equation (9) – presented entirely without justification.

Remarkably, page 5 also alludes to an explicit aim "to provide...a fully mechanistic description..." But no current fire-vegetation model, including this one, could plausibly be described as 'fully mechanistic.' The authors seem to admit this later (page 10), when they describe the treatment of human ignitions in their model as "very simplistic".

In principle there is always room for new models. The one presented here is new, in

that it differs in several respects from earlier models, including Thonicke et al. (2010). However, to be worth publishing, a new model should surely represent an identifiable advance, in terms of either derivation or performance, over existing models. For SEVER-FIRE, as presented, this seems not to be the case.

References not cited in the preprint

Baudena, M., S.C. Dekker, P.M. van Bodegon, B. Cuesta, S.I. Higgins, V. Lehsten, C.H. Reick, M. Rietkerk, S. Scheiter, Z. Yin, M.A. Zavala and V. Brovkin (2015) Forests, savannas, and grasslands: bridging the knowledge gap between ecology and Dynamic Global Vegetation Models. Biogeosciences 12: 1833-1848.

Kelley, D.I., I.C. Prentice and S.P. Harrison (2014) Improved simulation of fire-vegetation interactions in the Land surface Processes and eXchanges Dynamic Global Vegetation Model (LPX-Mv1). Geoscientific Model Development 7: 2411-2433.

Lasslop, G., K. Thonicke and S. Kloster (2014) SPITFIRE within the MPI Earth system model: Model development and evaluation. Journal of Advances in Modeling Earth Systems 6: 740-755.

Pfeiffer, M., A. Spessa and J.O. Kaplan (2013) A model for global biomass burning in preindustrial time: LPJ-LMfire (v1.0). Geoscientific Model Development 6: 643-685.

Prentice, I.C., D.I. Kelley, S.P. Harrison, P.J. Bartlein, P.N. Foster and P. Friedlingstein (2011). Modeling fire and the terrestrial carbon balance. Global Biogeochemical Cycles 25: GB3005.

Rabin, S.S., J.R. Melton, G. Lasslop, D. Bachelet, M. Forrest, S. Hantson, J.O. Kaplan, F. Li, S. Mangeon, D.S. Ward, C. Yue, V.K. Arora, T. Hickler, S. Kloster, W. Knorr, L. Nieradzik, A. Spessa, G.A. Folberth, T. Sheehan, A. Voulgarakis, D.I. Kelley, I.C. Prentice, S. Sitch, S. Harrison and A. Arneth (2017) The Fire Modeling Intercomparison Project (FireMIP), phase 1: experimental and analytical protocols with detailed model descriptions. Geoscientific Model Development 10: 1175-1197.

Venevsky, S., Y. Le Page, J.M.C. Pereira and C. Wu (2018) Analysis fire patterns and drivers with a global SEVER-FIRE model incorporated into Dynamic Global Vegetation Model and satellite and on-ground observations. Geoscientific Model Development Discussions https://doi.org/10.5194/gmd-2018-178.

Wu, M., W. Knorr, K. Thonicke, G. Schurgers, A. Camio and A. Arneth (2015) Sensitivity of burned area in Europe to climate change, atmospheric $CO_2$ levels, and demography: A comparison of two fire-vegetation models. Journal of Geophysical Research – Biogeosciences 120: 2256-2272.

Yue, C., P. Ciais, P. Cadule, K. Thonicke, S. Archibald, B. Poulter, W.M. Hao, S. Hantson, F. Mouillot, P. Friedlingstein, F. Maignan and N. Viovy (2014) Modelling the role of fires in the terrestrial carbon balance by including SPITFIRE into the global vegetation model ORCHIDEE – Part 1: simulating historical global burned area and fire regimes. Geoscientific Model Development 7: 2747-2767.

---

## Short Comment (SC3) · 2 Oct 2018

I.Colin Prentice, a  prominent Earth system scientist, wrote a short comment to our manuscript with an extremely negative and (we contend) unjustified title "Limited progress in fire modelling". We feel the reviewer's comments are mostly in response / directed to the style of the written text, rather than any substantive critique of the science advances we present. We hope this negative view of our manuscript is a misunderstanding due to our writing style as we are non-native English speakers, and it was not our intention to offend any peers in the community. Nevertheless, firstly we want to thank I.Colin Prentice for his comments, which we believe will improve our manuscript, and help us avoid unnecessary and unintentional conflict among our peers.

We address individual comments below.

**Introduction sentence:**

"*The presentation of this model is marred by unfounded claims, inconsistencies, and unwarranted disparagement of earlier research.*"  No details are given later on what are these "*unfounded claims*" (except of critics of equation 9 (we will come later to it)) and/or "*inconsistences*". By "*unwarranted disparagement of earlier research*" I.Colin Prentice probably means a rather innocuous statement in our manuscript that "…(some) global fire models (including Thonicke et al., 2010) contain sets of rather complicated equations with variety of coefficients which is hard to obtain, unless satellite derived functions are used" (we will come to this point later). Otherwise, we are at a loss where our text has offended.

**Main body:**

"*the new fire component is based on Venevsky et al.'s earlier (2002) regional fire model, RegFirm. Thus, the new model is assembled from the same components as the LPJ-SPITFIRE model published by Thonicke et al. (2010), eight years previously.*"

We thank the reviewer for recognizing the first author's major contribution to large-scale fire modelling, i.e. how components of LPJ-SPITFIRE actually stem from RegFIRM of 2002. We contend that RegFIRM was actually pioneering work for process-oriented DGVM fire models and major components and many ideas (using of Nesterov Index for fire danger estimate, division to lightning and human fires, estimate of number of fires, elliptic form of area burnt and using Rothermel equation with fuel bulk density) were subsequently adopted by many currently used fire models, e.g. a majority of fire models used in FIREMIP inter-comparison project (see Figures 3 and 4 in Rabin et al., 2017) have the same composition as RegFIRM. Here we present the SEVER-FIRE model that builds on and advances these earlier pioneering developments), e.g. we improve earlier algorithms and introduce new functionality with respect: 1) to estimate the numbers of lightning fires from data on convective activity in the atmosphere 2) to estimate numbers of human fires from urban against rural population (timing of their appeareance in natural landscapes and their ratio) and regional wealth index, as well 3) to estimate more realistically fire duration, which in the new model depends on human suppression and weather situation and can last for several days. All represent significant new developments.

"*The preprint does not cite more recent developments and applications based on SPITFIRE, such as Prentice et al. (2011), Pfeiffer et al.(2013), Lasslop et al. (2014), Kelley et al. (2014), Yue et al. (2014), Baudena et al.(2015) and Wu et al. (2015)*".

We agree with critics and certainly will include and cite all references suggested by the reviewer.

"*The preprint does not claim any superior ability to reproduce observed patterns. It does however claim to be based on a different, and implicitly superior, approach to modelling fire.*"

This is a clear misunderstanding as we definitely do not claim that our approach to modelling of fire is implicitly superior; we simply contend that our work is a valuable contribution to the field. Our approach is arguably 1) more suitable for climate change future and past studies because, where input satellite data is unavailable, it 2) allows use of abundant historical national fire statistics not limited to the last few decades and 3) allows to validate fire model against historical regional number of lightning/number of human fires. This is described in lines 19-28 at page 3 of our preprint.

"*For example, page 3 refers to other models (including Thonicke et al., 2010) being based on "sets of rather complicated equations with variety of coefficients (despite they name themselves intermediate complexity models)...*"

We apologise for any offense caused, this was certainly not our intention. This was a poorly formulated sentence and negative message. We simply wanted to highlight a general and major challenge in large-scale modelling which is to strike a balance between mechanistic detail and availability of data, i.e. adding complexity does not necessary improve a model as it adds to the degrees of freedom, and thus with insufficient data, will not reduce uncertainty.

First author Venevsky respectfully agrees to disagree with the reviewer on the merit of some changes made going from RegFIRM to LPJ-SPITFIRE, as has been documented in the past correspondence in the year 2010 ([https://www.biogeosciences-discuss.net/7/C331/2010/bgd-7-C331-2010-print.pdf](https://www.biogeosciences-discuss.net/7/C331/2010/bgd-7-C331-2010-print.pdf)). However this is a distraction in the context of our new study, where we wish to showcase our recent advances with our new SEVER-FIRE model. We therefore delete the sentence.

"*Later, the preprint states about SEVER-FIRE that "No satellite derived data are used as an input of the model. Only physically based or just 'common sense' based equations from on-ground observations allow direct implementation of SEVER-FIRE model..." and "Unlike in other global DGVM fire modules ... all equations are kept simple following ideology of Reg-FIRM."*"

We agree in the second sentence that "common sense" based equations" is a bad expression. This is a terminology mistake and we meant actually "statistically-based equations from on-ground observations". As mentioned above there is a challenge in large-scale fire models, to keep equations relatively simple to avoid additional multiple unknown / uncertain parameters. We try to justify the merit of our new work in this context, and differentiate it from some earlier modelling efforts. However, the reviewer may have taken exception to this comment, therefore we remove "Unlike in other global DGVM fire modules" to keep the focus entirely on SEVER-FIRE and keeping the strategy of Reg-FIRM.

"*Thus, earlier models are criticized for their use of satellite data as input (for lightning frequency in the case of SPITFIRE) although the basis for this objection is not stated*"

We argue that it would be advantageous if one can produce long-term fire relationships without depending on remote-sensing, which is available for a relatively short period of time (a few decades). Fire return intervals can be of the order of hundreds of years, whereas remote sensing is available for several decades. Therefore using remote sensing to derive relationships implicitly assumes a space for

time substitution, which may or may not hold. Also our approach in turn allows the remote sensing to be employed as a valuable evaluation dataset, albeit over this limited time interval.

In fact, the problems with using satellite data as input are also mentioned in one of the latest studies for Africa, a study recommended by the reviewer, namely of Baudena et al,, 2015, where frequency of fire was prescribed as an input based on MODIS data: "LPJ-GUESS-SPITFIRE simulation results do not show any low tree cover value (e.g., below 50% cover) for rainfall higher than about 900mm/yr. In other words, this model (quite surprisingly) does not predict any savanna in mesic environments….this issue is also likely to be connected to fire intensity depending on fuel moisture. In this model, fire occurrence in a patch is calculated probabilistically from the proportion of burned area as determined from the remote sensing product. This probabilistic approach is necessary because the temporal extent of the remote sensed data (now only ca. 10 years), used to generate the probability of burned area for each pixel, is much shorter than the extent of the climate data for which the model was run (ca. 100 years)." We are going to include this example of negative influence of RS based input for result of DGVM based fire model: **For example, use of remote sensing derived fire frequency for Africa as an input to SPITFIRE for Africa, resulted in absence of savanna for the area with annual rainfall larger then 900 mm/yr (Baudena et al., 2015). This shortcoming of process-oriented fire model is attributed by authors to the short temporal extent of initial remote sensed data used for preparation of input data.**

"*Appeals are made to simplicity and 'common sense' – the former being a defensible aim, the latter not a scientific concept – and, curiously, to an 'ideology'*."

As mentioned earlier we made the terminological mistake, using expression "common sense" equations" and now change it "**statistically-based equations from on-ground observations**". So, false impression of "appeal to common sense" related only to this mistake. However we repeat our approach is one of simplicity, to avoid the issues of expanding the degrees of freedom and potentially adding to uncertainly, as we have aluded to above, and this is our strategy or ideology. We replace the word "ideology" with "**strategy**".

Actual justification of our approach to fire modelling (changed in accordance to the reviewer's comment) is described in lines 32 on page 3 to line 16 of page 4, ( sorry for lengthy self citation): "SEVER-FIRE (Socio-Economic and natural Vegetation ExpeRimental global fire model is incorporated into the SEVER_DGVM (Venevsky and Maksyutov, 2007; Wu et al., 2017), which is a modification of LPJ-DGVM (Sitch et al., 2003) for daily time step computation. SEVER-FIRE model is a follow up of Reg-FIRM and is designed using principles of the parent model. No satellite derived data are used as an input of the model. Only physically based or just statistically based equations from on-ground observations allow direct implementation of SEVER-FIRE model in any DGVM or ESM for investigation of **future** global change impacts or **past** global fire regimes reconstruction. One of the major focuses of SEVER-FIRE model is an implementation of pyrogenic behaviour of humans (timing of their activities and their willingness/necessity to ignite or supress fire), related to socio-economic and demographic conditions in a geographical domain of the model application. Importance of description of pyrogenic behaviour of humans are confirmed by recent findings of bi-modal fire regimes, reflecting human fingerprint in global fires dynamics (Benali et al., 2017), as well as by differences in timing of ignitions determined by religious background in Sub-Sahara Africa (Pereira et al., 2015). Fire weather regimes, set by climate dynamics, and fuel state set by vegetation dynamics are other important drivers in SEVER-FIRE model. SEVER-DGVM fire module, based on climate observations, external anthropogenic parameters, and SEVER-DGVM derived vegetation, estimates fire incidence and emissions. The resulting vegetation disturbance feeds back to the DGVM, ensuring a fully coupled system"

We intent to build a process-oriented fire model which will allow simulation both in the future and in the past, which will not be limited by time span of RS input (twenty years) and which will use for validation not only limited in time RS based area burnt product but also historical statistics on number of lightning and human fires and which will describe physically reasonably socio-economic, vegetation and climate driving of fire regimes.

"*These appeals do not amount to a convincing case for a new model, especially as it includes equations – notably those determining rates of human ignition, e.g. equation (9) – presented entirely without justification*"

The only substantive technical comment, and we found it confusing …Rate of human ignitions is actually determined in equation 8, not 9. This one (eq. 8) is taken (as it explained in RegFIRM paper in 2002) directly (but with slightly modification) from calculation methods of Russian Forest Service, which are in practical use for almost 50 years. Equation 9 describes "Mathematical expectation of number of ignition produced by one person for millions of hectares *a*" Estimation of *a* for modest high income region (Spain) and low income region (Sahel) is described in RegFIRM 2002 paper. Several other values of *a* are known for other regions from methods of Russian Forest Service (for European Russia, Asian Russia, districts of Russia). These known and estimated values of *a* were logarithmically fitted to wealth index WI of UN Human Settlement Program. Justification of variation of *a* value is described in lines 23 -28 page 9 of the preprint. We are going to add details from this answer in the next version of the preprint in order to underline substantial step done in development of RegFIRM with fixed mathematical expectation of number of ignition produced by one person for millions of hectares  to spatially distributed variable determined by wealth status of population.

"*Remarkably, page 5 also alludes to an explicit aim "to provide...a fully mechanistic description..." But no current fire-vegetation model, including this one, could plausibly be described as 'fully mechanistic.' The authors seem to admit this later (page 10), when they describe the treatment of human ignitions in their model as "very simplistic".*"

Very simplistic model in ecology can be also fully mechanistic, e.g. Lotka-Volterra equations for prey and predator number of individuals.

In general, this comment forwards us to some unnecessary terminological discussion. The first opened page in the Internet states that:

This is a rather minor philosophical argument. We remove the word "fully" to avoid conflict with the reviewer, and replace with "comprehensive", as we maintain that SEVER-FIRE includes key individual parts of fire phenomena, e.g. first major individual part of SEVER-FIRE is working. The second major individual part (estimate of areas burnt), based on the first part, is also working as demonstrated by comparison with RS data. The third major individual part (estimate of fire carbon emission) is also OK as seen from RS data. Thus, we think that SEVER-FIRE is **(a first) step to comprehensive mechanistic fire model**. This is how we will write in our next version of the preprint.

**Conclusion**

"*In principle there is always room for new models. The one presented here is new, in that it differs in several respects from earlier models, including Thonicke et al. (2010).*"

Very positive conclusion. Indeed, SPITFIRE and SEVER-FIRE differ principally in all but first component (estimate of Nesterov based fire danger Index) of the fire model.

Here is the list of significant changes which we have undertalen to move RegFIRM to SEVER-FIRE.

Number of lightning fires **simulation**: RegFIRM fixed value

> SEVER-FIRE based on convective activity in the atmosphere

Number of lightning fires **validation**: RegFIRM not done

> SEVER-FIRE done for limited area

Number of human fires **simulation**: RegFIRM based on equations of Russian Forest Service

> SEVER-FIRE based modified equations of RegFIRM with subdivision of population to urban and rural and different timing of contact with natural vegetation for these two types and with considering wealth status of population

Number of human fires **validation:** RegFIRM done for Spain

> SEVER-FIRE done for Canada and large areas in Canada, for Spain (as in RegFIRM)

Areas burnt **simulation**: RegFIRM equations based on Rothermel and eliptic up-wind form of area burnt perimeter, termination by natural reason – rain, maximum fire duration s

SEVER-FIRE initial equations of RegFIRM, termination by natural reason – rain and by human suppression, depending from geographical location of large human settlements

Areas burnt **validation:** RegFIRM Hystrorical statistics for Spain areas burnt is used

> SEVER-FIRE global RS areas burnt product is used

To conclude **BOTH DERIVATION AND VALIDATION OF RegFIRM AND SEVER-FIRE ARE DIFFERENT.**

"*However, to be worth publishing, a new model should surely represent an identifiable advance, in terms of either derivation or performance, over existing models. For SEVER-FIRE, as presented, this seems not to be the case*"

Final conclusion of I.Colin Prentice contradicts the first sentence of his conclusion section where he acknowledges novelty of SEVER-FIRE.

Identifiable advance of SEVER-FIRE (Socio-Economic and natural Vegetation ExpeRimental global fire model) in terms of derivation for fire modelling is shown (as we already discussed) in lines 32 on page 3 to line 16 of page 4 of the preprint.

We cannot state performance advance of SEVER-FIRE model over other DGVM fire models unless we compare the models with the same input and comparison data base protocols. However, we hope that new approach undertaken by SEVER-FIRE for socio-economic driving of global fire regimes and for

description of lightning fires can provide us with better reproduction of reality in the past, present and thus give added confidence in our future projections.

Sergey Venevsky

---

## Short Comment (SC4) · 2 Oct 2018

Thank you for your comment, we will place all necessary information on code availability and the model version in the revised version of the preprint

---

## Referee Comment (RC1) · Anonymous Referee #1 · 19 Oct 2018

**Analysis fire patterns and drivers with a global SEVER-FIRE model incorporated into Dynamic Global Vegetation Model and satellite and on-ground observations**

**Sergey Venevsky et al.**

The paper presents a dynamical fire model coupled with a vegetation scheme that is the global extension of a previous regional version designed for the Iberian peninsula. In general the topic is quite hot in the community. There is consensus that for many aspects we should go toward integrating more and more processes into the Earth system modelling, as it is proved that one process can improve the predictability of even not affected variables.

While the topic and all associated developments are very welcome I found that the paper does not live up to my expectations. First of all I should agree with the previous comment that the introduction feels more like a rant about others work that a fair assessment of the quality of the presented model. Moreover most of the time there is no scientific justification on why other approaches appear to be inferior. I do not find that the use "of rather complicated equations" (line 16) could be considered as an objective metric to judge the (non)- quality of a model in my opinion

But what annoys most is the conveyed idea that satellite data and their use is almost inherently wrong and/or inferior to local measurements. This is just a personal thinking of the authors contradicted in large part by the tangible improvements that satellite data have brought to many communities including oceanography, numerical weather prediction and obviously fire mapping. Clearly there are limitations in satellite data but so there are in using local observations or even fire lab experiments as the representativeness is a serious issue there.

In my opinion statements like "No satellite derived data are used as an input of the model. Only physically based or just 'common sense' based equations from on-ground observations allow direct implementation of SEVER-FIREModel..." should be removed as they have no quality justification apart from the liking of the authors.

*Essentially I highly recommend to rewrite the introduction removing all the assertion that cannot be justified scientifically and highlighting the innovative aspect of the model proposed.*

Methods

The description of the model is a bit chaotic possibly due to the fact that a big part of the model had already been developed, therefore equations seems to appear out of no-where.
I understand the need to describe the model but if modelling components were fully described somewhere else than a reference to a previous publication should suffice. Specific points:

1. the fire danger index is a byproduct of the model and not a model component and should be put later.
2. In the equation (7) for the number of expected fire from lightning I was expecting to have a soil moisture component as that would discriminate between wet and dry lightning. I believe the parameter moist is a constant ? or is this soil moisture? Please clarify.
3. The parametric equations (5) and (6) need some justification are these teh fit over some data ? Is this published somewhere else? If so they should be removed and the paper should only concentrate on what is new in this model.
4. In the analysis in figure 2, how you make sure the fire were ignited from a lightning ? Equation 9. I wonder how you set the parameter a. Why did you decided that 1 fire over millions of hectares is a reasonable number? Also what is it millions of hectares? 1,2 ,10 ?
5. I suppose equations 9 and 10 have been derived somewhere else ? as all appears pretty cripticat
6. The need for a simplification when considering human induced fires is understandable. One thing I would   add is a fire management factor. So in Europe it is not just a matter of GDP- or wealth but also of controlling program in place.
7. Page 11 line 5 you mean EFFIS? Suppression makes sense in Europe and 2 days is probably reasonable. However there are many places were suppression does not take place. Is this a global parameter ? Please comment on this

Data

1.  Please specify if the  GFED dataset used include small fires.

Results

Results are difficult to judge as the datasets used for validations are affected themselves from large uncertainties. The model seems to produce reasonable spatial patterns for burned areas and a good improvements in the burning  temporal variability especially when large anomalous conditions take place as the ones induced by ENSO. I do not see the lack of a big improvements as a problem as this  is a first overall assessment of the global  model and components can be tuned and improved if a specific aspects is proven very relevant for the fire process.

Final remark

The paper is very dishomogeneous in the way is written. The discussion for example is very nicely worded while the method session is badly explained and difficult to follow as many equations are just taken out without clearly stating if this is the outcome of a previous analysis (I suspect that is the case). A throughout re-writing of the introduction is a strong requirement as at the moment, a part from upsetting an entire community,  is not making a good service to the model either as does not explain what are the innovative aspects
Finally as this is the presentation of a new model or  at least a substantial development of an old one, I would suggest to extend the "code availability" section giving more details about the model itself (programming language, input output format/ licence etc etc)

---

## Short Comment (SC5) · 27 Oct 2018

Sergey Venevsky has responded in detail to my commentary. He states correctly that my comments are principally about the way in which the new model was presented, rather than being a critique of specific aspects. In his response he has clarified a number of points that were not clear to me on reading the preprint, and suggested various ways in which the presentation could be improved. In particular, he has clarified what are considered to be the most important innovations introduced in the model. Inclusion of these changes will improve the manuscript.

---

## Referee Comment (RC2) · Anonymous Referee #2 · 30 Oct 2018

**Review summary**

In this manuscript, Venevsky et al. describe a new fire module, SEVER-FIRE, incorporated into the SEVER dynamic global vegetation model (DGVM). SEVER-FIRE is largely based on the Reg-FIRM fire model, for whose description Venevsky was also lead author, and which provided the structural foundation upon which many modern global fire models have been built. SEVER-FIRE includes several new elements relating to fire ignition (by both lightning and humans) and fire termination, which seem

likely to improve model realism.

Many different approaches have been used in various aspects of global fire modeling, and the new elements introduced in this manuscript are welcome as alternative mechanisms and parameterizations. A new global fire-vegetation model, moreover, could add weight to efforts to explore the uncertainty related to fire drivers and the future of fire regimes around the world. For that reason, I think this manuscript could represent an important contribution to the fire modeling literature.

That said, I recommend that the manuscript be **resubmitted with major revisions**. My explanation follows.

**Main critique**

Previous comments on this manuscript have highlighted the tone of parts of the paper as problematic. While I don't see it as overly hostile, I do agree that revisions should be made in the aim of reflecting the authors' respect for previous work. In their reply to Colin Prentice's comment, the authors have indicated that they intend to make changes in that direction, so I will leave aside questions of tone and language.

I do have some concerns regarding the content of the discussion, however. The modeling approach of Venevsky et al. is to minimize the use of parameterizations based on remote sensing (here, "remote-sensing approach") and to instead favor mechanisms and relationships derived from first principles or laboratory-scale experiments ("first-principles approach"). This, they assert, may confer an advantage because their parameterizations may hold true far into the past or future (i.e., outside the satellite era) where remote-sensing-derived parameterizations do not. I can agree with that to some extent, in principle. However, Venevsky et al.—in the

original manuscript and in their reply to Prentice—need to rethink how they discuss this.

In their reply to Prentice, the authors cite Baudena et al. (2015) as supporting their contention that including parameterizations based on remote sensing data can result in unreliable models. Specifically, they quote this passage (quoted here in full):

> LPJ-GUESS-SPITFIRE simulation results do not show any low tree cover value (e.g., below 50 % cover) for rainfall higher than about 900 mm yr$^{-1}$ (Fig. 2b). In other words, this model (quite surprisingly) does not predict any savanna in mesic environments. In the model, though fire frequency is prescribed from the satellite data, fire spread depends on fuel load (Fig. 3c) and fuel moisture, and thus unfavorable conditions might still prevent fires. Both grass and tree presence increases fire intensity, opening up space, and thus favoring grasses. This is not strictly a positive grass–fire feedback because grass-free areas can also burn. Thus, as grasses are not fostered by the positive feedback with fire, they are always outcompeted by trees in LPJ-GUESS-SPITFIRE when water availability is high, and they do not survive above approximately 900 mm yr$^{-1}$. At the same time, this issue is also likely to be connected to fire intensity depending on fuel moisture. In this model, fire occurrence in a patch is calculated probabilistically from the proportion of burned area as determined from the remote sensing product. If fire occurs in a period of high fuel moisture, the intensity will be limited, thus having little effect on vegetation. This probabilistic approach is necessary because the temporal extent of the remote sensed data (now only ca. 10 years), used to generate the probability of burned area for each pixel, is much shorter than the extent of the climate data for which the model was run (ca. 100 years).

And here is the authors' interpretation, which they intend to include in their revision:

For example, use of remote sensing derived fire frequency for Africa as an input to SPITFIRE for Africa, resulted in absence of savanna for the area with annual rainfall larger then 900 mm/yr (Baudena et al., 2015). This shortcoming of process-oriented fire model is attributed by authors to the short temporal extent of initial remote sensed data used for preparation of input data.

That is unfortunately a misinterpretation of the Baudena et al. (2015) text. As described in Thonicke et al. (2010), Lehsten et al. (2009, 2016), and Rabin et al. (2017)—and as Venevsky et al. know, given their familiarity with how relevant parts of SPITFIRE were derived from Reg-FIRM—SPITFIRE does of course have a module that, just as with SEVER-FIRE, endogenously computes fire occurrence. In Baudena et al. (2015), that module in LPJ-GUESS-SPITFIRE (and two other global fire-vegetation models) was experimentally disabled and replaced with exogenous, remotely sensed burned area, with the goal of isolating and comparing the fire-vegetation models' representation of fire's ecological effects rather than fire occurrence and spread. In the quoted text, Baudena et al. (2015) are attributing the poor performance of LPJ-GUESS-SPITFIRE not to the use of satellite data (which Baudena et al. effectively consider a true representation of reality) but rather to LPJ-GUESS-SPITFIRE not representing fuel availability and moisture in a realistic way. The relevant mechanisms in LPJ-GUESS-SPITFIRE were not derived from remote sensing data.

Venevsky et al. also, in their reply to Prentice, suggest that the Baudena et al. (2015) example shows a disadvantage of the remote sensing approach in the present as well. However, the example does not support their case:

- It is the result of a contrived experiment that does not reflect how most global fire-vegetation models actually work.

- The only global fire-vegetation model I can think of that does directly input

satellite-derived burned area (LM3-FINAL.1; Rabin et al., 2018) would not be negatively affected by that input in the present. This is because LM3-FINAL.1 (a) only applies those burned areas on cropland and pasture, thus avoiding the problem with bad fire inputs leading to bad community composition, and (b) uses constant combustion completeness and fractional mortality factors that would not be affected by fire occurring on wet vs. dry days. Rabin et al. (2018) do acknowledge that the use of this input is problematic when applied outside the period of its derivation.

As I've said, I agree with the authors that a first-principles approach could be advantageous because it seems more likely to result in parameterizations that are more robust outside the satellite era, but I cannot think of how any example using historical data would support their case. Instead, I think the best thing the authors could write is what they wrote in their reply to Prentice:

> We argue that it would be advantageous if one can produce long-term fire relationships without depending on remote-sensing, which is available for a relatively short period of time (a few decades). Fire return intervals can be of the order of hundreds of years, whereas remote sensing is available for several decades. Therefore using remote sensing to derive relationships implicitly assumes a space for time substitution, which may or may not hold. Also our approach in turn allows the remote sensing to be employed as a valuable evaluation dataset, albeit over this limited time interval.

However, I am actually not convinced that SEVER-FIRE even *is* more grounded in first principles than most other global fire-vegetation models! I see at least one instance where remote sensing or other large-scale, recent historical datasets have been used:

- Equations 1–6, governing lightning ignitions, were derived from national networks

of ground-based sensors in the United States and Canada in 1997 (Allen & Pickering, 2002).

- Equation 9 may also have used such a dataset, although it's not clear exactly how it was parameterized. In their reply to Prentice, the authors mention that the value of $\bar{a}$ for peninsular Spain was derived in the Reg-FIRM description (Venevsky et al., 2002); while I was not able to totally follow the chain of logic presented there, I do understand generally the strategy. However, I do not see the parameterization for the Sahel that, according to the authors' reply to Prentice, is also supposedly in Venevsky et al. (2002). More importantly, even in their reply to Prentice, the authors do not describe what historical fire occurrence data they used to derive Equation 9. Was it satellite data? If so, that undermines the authors' insistence that SEVER-FIRE has an advantage due to independence from parameterizations based on remote sensing data. Or was it instead based on national statistical databases? There are issues with those as well:

  - They only exist in certain wealthy countries.
  - They may not be reliable going back into the mid-20th century.
  - They depend to some extent on the satellite record for recent decades.
  - It would still be basing a part of the model on some external data which, although based on a longer time period than the satellite record, could still fail to be representative of mechanisms far in the past or future.

This is not to say that SEVER-FIRE is an outlier; essentially all global fire-vegetation models are designed to reproduce a limited time series of historical data, either through explicit parameterization processes or through manual model tuning. Global fire models are typically classified into two groupings—purely empirical models and quasi-mechanistic models—which differ in their reliance on parameterizations derived from historical data. See, for example:

- The anthropogenic ignition components of (most of) the eight models included in Table S1 in the Supplement of Rabin et al. (2017)

- The parameter estimation (using the Levenberg-Marquardt algorithm) described for the quasi-mechanistic FINAL.1 in Rabin et al. (2018)

- Purely empirical models such as SIMFIRE (Knorr et al., 2014, 2016)

Thus SEVER-FIRE, rather than being categorically different from most other global fire-vegetation models (a "purely mechanistic" model, perhaps) as Venevsky et al. contend, seems instead to be more first-principles-based only by a matter of degree (i.e., it derives lightning flash rate from weather rather than from a historical-derived climatology, although that derivation does itself depend in part on historical data).

Finally, I agree with Reviewer 1 that the satellite record is not unique in its susceptibility to non-representativeness. Even completely accurate, decades-long, ground-based measurements could only be assumed to be representative of the time period covered, with whatever plant species, climate/weather patterns, and anthropogenic activity was there at the time. And of course such records are not completely—or even consistently—accurate anyway! Furthermore, such records are not global in coverage, so even though the problem with space-for-time substitution is lessened relative to the satellite record (not eliminated completely), a *space-for-space* problem is worsened. Likewise, laboratory-based experiments, such as those regarding the ignition efficiency of lightning strikes, depend on the species of plant litter involved—even an experiment sampling a wide variety of plant species from across the planet could fail to be representative of species far into the past or future. The brief temporal coverage of the satellite record may make it especially vulnerable to failures of robustness, but other datasets have their own problems.

Every development team has their own principles that they bring to model construction. If those principles represent a significant break with the dominant mode of thinking in the field, it makes sense to spend time in the model description discussing them. However, Venevsky et al. seem to have a perfectly normal quasi-mechanistic fire model in SEVER-FIRE. Thus, this manuscript should be rewritten to focus on the model itself (especially where it differs from previous models) rather than the philosophy that governed its design.

**Other major comments**

1. Apparent from the comments of Prentice and Reviewer 1, as well as my read of the manuscript, is that the authors need to improve the Introduction, Methods, and Discussion sections to better highlight the novel aspects of SEVER-FIRE.

2. When explaining novel parts of SEVER-FIRE, the derivation process should always be fully explained—as the authors did for their equations regarding lightning strikes. Such explanation needs to be added for:

   - The wealth dependence of anthropogenic ignitions (Eq. 9; as they mention they will do in their reply to Prentice)
   - The limitation of fire duration to two days. This limitation may have contributed to SEVER-FIRE's underestimation of burned area in the boreal region: Korovin (1996) found that almost 70% of the burned forest in Russia over 1947–1992 resulted from fires that burned for more than *ten* days.

3. The factor $timing_j$, which modulates the frequency of human ignitions depending on the time of year, seems rather ad-hoc but could nevertheless be of use for many fire models. The authors should demonstrate that including it actually improves the simulation of annual total and/or seasonal timing of burned area.

[Figure]

4. A glaring hole in many global fire models is that they do not allow multi-day burning, and so SEVER-FIRE's inclusion of this is most welcome. However, as with $timing_j$, the authors should demonstrate that including this parameterization improves their model.

5. I disagree with Reviewer 1's critique that the paper should be condensed by removing previously-published model components and instead directing readers to those publications. It is too easy to gloss over important differences that may have arisen in the time since the original publication, and makes it too difficult for the reader to learn about the model. One alternative could be to move explanation of non-novel model parts to one or more Appendices (or, less preferably in my opinion, a separate Supplement). The authors should also consider constructing a table-based description of their model to match the form of the supplementary tables in Rabin et al. (2017). This would enable a much simpler comparison between SEVER-FIRE and the models described there, and would ensure a complete description of all relevant aspects of the model.

6. The authors should explain why the model outputs were compared to GFED2, instead of the more recent GFED3(s) or GFED4(s), which would have a number of advantages:

   - These datasets cover nearly twice the time period as GFED2, which would increase the time period available for comparison—which the authors acknowledge as a weakness.
   - GFED3 incorporated an improved burned area detection algorithm (Giglio et al., 2010).
   - GFED4 incorporated further improvements to the burned area detection algorithm (Giglio et al., 2013).
   - The "s" versions of GFED3 and GFED4 are boosted by burned area estimated for small fires that the original algorithms fail to detect (Randerson et

al., 2012).

**Minor comments and technical corrections**

1. P10 L24–25: This sentence should cite the "other global fire models," as well as perhaps Rabin et al. (2017), which provides a comprehensive overview and comparison of a number of global fire models.

2. P11 L15 (Eq. 12): This equation structure does not seem to account for the fact that, for a given rate of linear spread, an older fire has a longer fireline and thus will add more burned area per unit time than a more recent fire. This could be a contributing factor to the underestimation of burned area in boreal regions, where large, long-lasting fires contribute significantly to total burned area. I do not consider this a critical issue, but it's something the authors should definitely mention.

3. P12 L17–26 (Sect. 2.2.2): It would be nice to see, probably in a Supplement, figures showing the input data described here.

4. P13 L18: "As a DGVM" should be deleted—there are certainly DGVMs that have the capability to output results that reflect the vegetated area in a gridcell. The authors should also explain (a) why they found it necessary to adjust the GFED data, rather than simply adjusting the SEVER outputs, and (b) what the net impacts of their adjustments were on global burned area.

5. P14 L19–21: A citation of Lasslop et al. (2015) should be made here.

6. P16 L14–16: This text implies that the overestimation of fire in India may have something to do with the fact that the model simulates grass there. In reality, it's

probably because of strong fire suppression resulting from high fractional coverage of cropland.

7. P18 L14: Mention should be made of the fact that these regions were originally created for use with GFED (Giglio et al., 2006).

8. Work is needed on the Discussion paragraph about anthropogenic impacts on fire (P21 L7–19):

   - Pfeiffer et al. (2013) should be mentioned, since they introduce a number of interesting ideas for modeling of human fire use.
   - "In Africa for example, the combination of a strong seasonal wet-dry climate with regular human ignitions favours high fire incidence." This sentence does not seem to fit with the idea introduced in the previous sentence; namely, that land use and agricultural practices are likely more directly related to fire incidence than wealth in certain regions.

**Works cited in this review**

Allen, D. J., & Pickering, K. E. (2002). Evaluation of lightning flash rate parameterizations for use in a global chemical transport model. Journal of Geophysical Research-Atmospheres, 107(D23), ACH 15–1–ACH 15–21. doi:10.1029/2002JD002066

Baudena, M., Dekker, S. C., van Bodegom, P. M., Cuesta, B., Higgins, S. I., Lehsten, V., et al. (2015). Forests, savannas, and grasslands: bridging the knowledge gap between ecology and Dynamic Global Vegetation Models. Biogeosciences, 12(6), 1833–1848. doi:10.5194/bg-12-1833-2015.

Giglio, L., van der Werf, G. R., Randerson, J. T., Collatz, G., & Kasibhatla, P.

(2006). Global estimation of burned area using MODIS active fire observations. Atmospheric Chemistry and Physics, 6, 957–974.

Giglio, L., Randerson, J. T., van der Werf, G. R., Kasibhatla, P., Collatz, G., Morton, D. C., & DeFries, R. (2010). Assessing variability and long-term trends in burned area by merging multiple satellite fire products. Biogeosciences, 7(3), 1171–1186.

Giglio, L., Randerson, J. T., & van der Werf, G. R. (2013). Analysis of daily, monthly, and annual burned area using the fourth-generation global fire emissions database (GFED4). Journal of Geophysical Research: Biogeosciences, 118(1), 317–328. doi:10.1002/jgrg.20042

Knorr, W., Kaminski, T., Arneth, A., & Weber, U. (2014). Impact of human population density on fire frequency at the global scale. Biogeosciences, 11(4), 1085–1102. doi:10.5194/bg-11-1085-2014

Knorr, W., Jiang, L., & Arneth, A. (2016). Climate, CO2, and demographic impacts on global wildfire emissions. Biogeosciences, 13, 267–282. doi:10.5194/bgd-12-15011-2015

Korovin, G. N. (1996). Analysis of the distribution of forest fires in Russia. In J. G. Goldammer & V. V. Furyaev (Eds.), Fire in Ecosystems of Boreal Eurasia (pp. 112–128). Dordrecht, The Netherlands.

Lasslop, G., Hantson, S., & Kloster, S. (2015). Influence of wind speed on the global variability of burned fraction: a global fire model's perspective. International Journal of Wildland Fire, 22, 959–969. doi:10.1071/WF15052
Lehsten, V., Tansey, K. J., Balzter, H., Thonicke, K., Spessa, A., Weber, U., et al. (2009). Estimating carbon emissions from African wildfires. Biogeosciences, 6, 349–360.

Lehsten, V., Arneth, A., Spessa, A., Thonicke, K., & Moustakas, A. (2016). The effect of fire on tree–grass coexistence in savannas: a simulation study, 25(2), 137–146. doi:10.1071/WF14205

Pfeiffer, M., Spessa, A., & Kaplan, J. O. (2013). A model for global biomass burning in preindustrial time: LPJ-LMfire (v1.0). Geoscientific Model Development, 6(3), 643–685. doi:10.5194/gmd-6-643-2013

Rabin, S. S., Melton, J. R., Lasslop, G., Bachelet, D., Forrest, M., Hantson, S., et al. (2017). The Fire Modeling Intercomparison Project (FireMIP), phase 1: experimental and analytical protocols with detailed model descriptions. Geoscientific Model Development, 10(3), 1175–1197. doi:10.5194/gmd-10-1175-2017

Rabin, S. S., Ward, D. S., Malyshev, S. L., Magi, B. I., Shevliakova, E., & Pacala, S. W. (2018). A fire model with distinct crop, pasture, and non-agricultural burning: use of new data and a model-fitting algorithm for FINAL.1. Geoscientific Model Development, 11(2), 815–842. doi:10.5194/gmd-11-815-2018

Randerson, J. T., Chen, Y., van der Werf, G. R., Rogers, B. M., & Morton, D. C. (2012). Global burned area and biomass burning emissions from small fires. Journal of Geophysical Research, 117(G4), G04012. doi:10.1029/2012JG002128

Thonicke, K., Spessa, A., Prentice, I. C., Harrison, S. P., Dong, L., & Carmona-Moreno, C. (2010). The influence of vegetation, fire spread and fire behaviour on biomass burning and trace gas emissions: results from a process-based model.

Biogeosciences, 7(6), 1991–2011. doi:10.5194/bg-7-1991-2010

Venevsky, S., Thonicke, K., Sitch, S., & Cramer, W. (2002). Simulating fire regimes in human-dominated ecosystems: Iberian Peninsula case study. Global Change Biology, 8(10), 984–998.

---

## Author Comment (AC1) · 26 Nov 2018

**Author response to Anonymous Reviewer #1 on: "Analysis fire patterns and drivers with a global SEVER-FIRE model incorporated into Dynamic Global Vegetation Model and satellite and on-ground observations" by Sergey Venevsky et al.**

We appreciate the constructive comments from the reviewers. Reviewer comments are in black, our responses are in blue.

And a revised document highlighting the tracked changes we have made based on these comments are also provided after the end of the response.

**Reviewer 1#**

**The paper presents a dynamical fire model coupled with a vegetation scheme that is the global extension of a previous regional version designed for the Iberian peninsula. In general the topic is quite hot in the community. There is consensus that for many aspects we should go toward integrating more and more processes into the Earth system modelling, as it is proved that one process can improve the predictability of even not affected variables.**

**While the topic and all associated developments are very welcome I found that the paper does not live up to my expectations.**

Thank you for your comments. We completely changed all wordings in Introduction and somewhere Methods and Discussion and now hope we met expectations of Reviewer 1.

**First of all I should agree with the previous comment that the introduction feels more like a rant about others work that a fair assessment of the quality of the presented model. Moreover most of the time there is no scientific justification on why other approaches appear to be inferior. I do not find that the use "of rather complicated equations" (line 16) could be considered as an objective metric to judge the (non)- quality of a model in my opinion**

**But what annoys most is the conveyed idea that satellite data and their use is almost inherently wrong and/or inferior to local measurements. This is just a personal thinking of the authors contradicted in large part by the tangible improvements that satellite data have brought to many communities including oceanography, numerical weather prediction and obviously fire mapping. Clearly there are limitations in satellite data but so there are in using local observations or even fire lab experiments as the representativeness is a serious issue there**

**In my opinion statements like "No satellite derived data are used as an input of the model. Only physically based or just 'common sense' based**

**equations from on-ground observations allow direct implementation of SEVER-FIRE Model...”** should be removed as they have no quality justification apart from the liking of the authors.

*Essentially I highly recommend to rewrite the introduction removing all the assertion that cannot be justified scientifically and highlighting the innovative aspect of the model proposed.*

Accept. The Introduction is re-written. And we delete all the assertion that cannot be justified scientifically and especially, highlight the innovative aspect of the model proposed.

**Methods**

**The description of the model is a bit chaotic possibly due to the fact that a big part of the model had already been developed, therefore equations seems to appear out of no-where. I understand the need to describe the model but if modelling components were fully described somewhere else than a reference to a previous publication should suffice.**

Here we have a bit of contradictive requests of Reviewer 1 and Reviewer 2. We are more inclined that we should describe all model elements as Reviewer 2 suggested (in case model components were already described we do it briefly).

**Specific points:**

**1. the fire danger index is a byproduct of the model and not a model component and should be put later.**

Sorry, we kindly disagree that FDI is not a model component and keep description as it was.

**2. In the equation (7) for the number of expected fire from lightning I was expecting to have a soil moisture component as that would discriminate between wet and dry lightning. I believe the parameter moist is a constant? or is this soil moisture? Please clarify.**

Thank you for this comment. Term *moist* is volumetric soil moisture in 0.1 m of upper soil layer. This is now mentioned in paragraph 2.1.1.

**3. The parametric equations (5) and (6) need some justification are these teh fit over some data? Is this published somewhere else? If so they should be removed and the paper should only concentrate on what is new in this model.**

Yes. This is the fit of data over eight fuel types for two classes of ignitions for positive and negative flashes. Now it is mentioned in the section 2.1.4--2)

Simulation of lightning ignition events and number of lightning fires.

**4. In the analysis in figure 2, how you make sure the fire were ignited from a lightning ?**

Number of lightning fires for provinces is published in Wierzchowski et al. (2002). Now it is mentioned in the caption for Figure 2.

**Equation 9. I wonder how you set the parameter a. Why did you decided that 1 fire over millions of hectares is a reasonable number? Also what is it millions of hectares? 1,2 ,10 ?**

This was a misprint. It should be read "one million hectares". Using of not SI area units here appeared because of origin of equation 8, which is equivalent to equations 10 and 14 in Venevsky et al. (2002). Scaling parameter a=0.0001 km2/mln. hectar was initially set (but not mentioned as I found) in my Reg-FIRM model (Venevsky et al., 2002) to convert mln hectares to square kilometers, here it was slightly modified (divided by 8 kgC/m$^{-1}$ average fuel density). See changes below equation (8).

**5. I suppose equations 9 and 10 have been derived somewhere else? as all appears pretty cripticat**

Equation 10 is similar to equation 7 and (in our opinion) does not need clarification. Equation 8 is explained in more details now. Equation 9 was derived as logarithmic regression of historically observed number of human fires, see new text below Eq.(9):

*"Equation (9) was obtained using logarithmic regression from geographically distributed observed number of human fires (map of average over 1974-1994 annual number of human fires for Spain (Vazquez and Moreno, 1998), map of average over annual number of human fires by Canadian ecoregions 1961-1995 (Stocks et al., 2002) and map of average over annual number of total fires (assumed to be all human) by African countries 1981-1991 (Barbosa et al., 1999). No division to rural and urban population was assumed when deriving Eq. (9)"*

6. **The need for a simplification when considering human induced fires is understandable. One thing I would add is a fire management factor. So in Europe it is not just a matter of GDP- or wealth but also of controlling program in place.**

Indeed. We put this comment in Discussion and note it for future development of SEVER-FIRE.

*"Fire management factor should be added to the model in the regions where coordinated wildfire controlling program is in place (e.g. existence and actions*

*of European Commissions Emergency Response Coordination Centre in Europe(https://ec.europa.eu/echo/what-we-do/civil-protection/forest-fires_en)).”*

**7. Page 11 line 5 you mean EFFIS? Suppression makes sense in Europe and 2 days is probably reasonable. However there are many places were suppression does not take place. Is this a global parameter? Please comment on this**

Yes. Sorry for typo, these are EFFIS database. Application of 2 days as maximum for fire duration at global scale limits our model. We write now in the text:

*“However, the limitation of maximum fire duration to two days was set due to range in the fire duration of EFFIS database, which covers mainly European domain. Globally this limitation may be not valid for remote high latitude areas, but even in these regions mathematical expectation of fire duration will be close to one day (see Korovin (1996))”*

*In Discussion:*

*“Study and parameterization of fire duration in remote areas is necessary for improvement of area burnt calculation in these areas.”*

*In Introduction:*

*“(e.g., setting maximum time of fire to two days but this may be updated and modified in the future by introducing the latest global fire duration datasets (Andela et al., 2018)”*

**Data**

1. **Please specify if the GFED dataset used include small fires.**

No small fires are included. This is now mentioned in the section 2.3

**Results**

**Results are difficult to judge as the datasets used for validations are affected themselves from large uncertainties. The model seems to produce reasonable spatial patterns for burned areas and a good improvements in the burning temporal variability especially when large anomalous conditions take place as the ones induced by ENSO. I do not see the lack of a big improvements as a problem as this is a first overall assessment of the global model and components can be tuned and improved if a specific aspects is proven very relevant for the fire process.**

Agree and thank you for your good summary.

**Final remark**

**The paper is very dishomogeneous in the way is written. The discussion for example is very nicely worded while the method session is badly explained and difficult to follow as many equations are just taken out without clearly stating if this is the outcome of a previous analysis (I suspect that is the case).**

**A throughout re-writing of the introduction is a strong requirement as at the moment, a part from upsetting an entire community, is not making a good service to the model either as does not explain what are the innovative aspects**

Thank you very much for your very constructive suggestions. Based on your specific comments before, we have revised the paper very carefreely. Especially, we re-write the Introduction part, including many innovative aspects of SEVER-FIRE, and clarify the Methods and Discussion parts.

**Finally as this is the presentation of a new model or at least a substantial development of an old one, I would suggest to extend the "code availability" section giving more details about the model itself (programming language, input output format/ licence etc etc)**

This has been done in the new 'code availability' part.

**References cited in this response**

[revised manuscript text omitted]

---

## Author Comment (AC2) · 26 Nov 2018

**Author response to Anonymous Reviewer #2 on: "Analysis fire patterns and drivers with a global SEVER-FIRE model incorporated into Dynamic Global Vegetation Model and satellite and on-ground observations" by Sergey Venevsky et al.**

**We appreciate the constructive comments from the reviewers. Reviewer comments are in black, our responses are in blue.**

**And a revised document highlighting the tracked changes we have made based on these comments are also provided after the end of the response.**

**Referee #2**

**Review summary**

In this manuscript, Venevsky et al. describe a new fire module, SEVER-FIRE, incorporated into the SEVER dynamic global vegetation model (DGVM). SEVER-FIRE is largely based on the Reg-FIRM fire model, for whose description Venevsky was also lead author, and which provided the structural foundation upon which many modern global fire models have been built. SEVER-FIRE includes several new elements relating to fire ignition (by both lightning and humans) and fire termination, which seem likely to improve model realism. Many different approaches have been used in various aspects of global fire modeling, and the new elements introduced in this manuscript are welcome as alternative mechanisms and parameterizations. A new global fire-vegetation model, moreover, could add weight to efforts to explore the uncertainty related to fire drivers and the future of fire regimes around the world. For that reason, I think this manuscript could represent an important contribution to the fire modeling literature. That said, I recommend that the manuscript be resubmitted with major revisions. My explanation follows.

**Main critique**

Previous comments on this manuscript have highlighted the tone of parts of the paper as problematic. While I don't see it as overly hostile, I do agree that revisions should be made in the aim of reflecting the authors' respect for previous work. In their reply to Colin Prentice's comment, the authors have indicated that they intend to make changes in that direction, so I will leave aside questions of tone and language. I do have some concerns regarding the content of the discussion, however.

The modeling approach of Venevsky et al. is to minimize the use of parameterizations based on remote sensing (here, "remote-sensing approach") and to instead favor mechanisms and relationships derived from first principles or laboratory-scale experiments ("first-principles approach"). This, they assert, may confer an advantage because their

parameterizations may hold true far into the past or future (i.e., outside the satellite era) where remote-sensing-derived parameterizations do not. I can agree with that to some extent, in principle. However, Venevsky et al.—in the original manuscript and in their reply to Prentice—need to rethink how they discuss this.

The arguments both in Introduction and Discussion were changed to meet requests of Reviewer 2 and 1. Now manuscript is focused mainly on innovations presented by SEVER-FIRE in comparison with other models.

In their reply to Prentice, the authors cite Baudena et al. (2015) as supporting their contention that including parameterizations based on remote sensing data can result in unreliable models. Specifically, they quote this passage (quoted here in full):

> LPJ-GUESS-SPITFIRE simulation results do not show any low tree cover value (e.g., below 50 % cover) for rainfall higher than about 900 mm yr−1 (Fig. 2b). In other words, this model (quite surprisingly) does not predict any savanna in mesic environments. In the model, though fire frequency is prescribed from the satellite data, fire spread depends on fuel load (Fig. 3c) and fuel moisture, and thus unfavorable conditions might still prevent fires. Both grass and tree presence increases fire intensity, opening up space, and thus favoring grasses. This is not strictly a positive grass–fire feedback because grass-free areas can also burn. Thus, as grasses are not fostered by the positive feedback with fire, they are always outcompeted by trees in LPJ-GUESS-SPITFIRE when water availability is high, and they do not survive above approximately 900 mm yr−1. At the same time, this issue is also likely to be connected to fire intensity depending on fuel moisture. In this model, fire occurrence in a patch is calculated probabilistically from the proportion of burned area as determined from the remote sensing product. If fire occurs in a period of high fuel moisture, the intensity will be limited, thus having little effect on vegetation. This probabilistic approach is necessary because the temporal extent of the remote sensed data (now only ca. 10 years), used to generate the probability of burned area for each pixel, is much shorter than the extent of the climate data for which the model was run (ca. 100 years).

And here is the authors' interpretation, which they intend to include in their revision:

> For example, use of remote sensing derived fire frequency for Africa as an input to SPITFIRE for Africa, resulted in absence of savanna for the area with annual rainfall larger then 900 mm/yr (Baudena et al., 2015). This shortcoming of process-oriented fire model is

**attributed by authors to the short temporal extent of initial remote sensed data used for preparation of input data.**

**That is unfortunately a misinterpretation of the Baudena et al. (2015) text. As described in Thonicke et al. (2010), Lehsten et al. (2009, 2016), and Rabin et al. (2017)—and as Venevsky et al. know, given their familiarity with how relevant parts of SPITFIRE were derived from Reg-FIRM— SPITFIRE does of course have a module that, just as with SEVER-FIRE, endogenously computes fire occurrence. In Baudena et al. (2015), that module in LPJ-GUESS-SPITFIRE (and two other global fire-vegetation models) was experimentally disabled and replaced with exogenous, remotely sensed burned area, with the goal of isolating and comparing the fire-vegetation models' representation of fire's ecological effects rather than fire occurrence and spread. In the quoted text, Baudena et al. (2015) are attributing the poor performance of LPJ-GUESS-SPITFIRE not to the use of satellite data (which Baudena et al. effectively consider a true representation of reality) but rather to LPJ-GUESS-SPITFIRE not representing fuel availability and moisture in a realistic way. The relevant mechanisms in LPJ-GUESSSPITFIRE were not derived from remote sensing data.**

Thank you very much. Yes. Indeed, Baudena et al. (2015) are not speaking about using remote sensing data for parametrization of global fire model, but rather about using of remote sensing data as input (which is also written in suggested changes to the manuscript). What I meant in my response to I. Colin Prentice is that short period of prescribed from RS areas burnt in comparison with climate data range (100 years) for LPJ-GUESS-SPITFIRE made it necessary to use probabilistic method for fire occurrence in each singular patch. Use of probabilistic method could obscure resulting tree cover in a high fuel moisture patch where fire is randomly prescribed. However, I also think in with line of Reviewer 2, that to larger extent deficiencies in representation of resulting tree cover in Africa are mainly determined by deficiencies in representation of fire intensity by LPJ-GUESS-SPITFIRE. The text from our response to I. Colin Prentice is misleading in contexts of impact of RS data to parametrization, thus the text will be discarded in our final revision.

**Venevsky et al. also, in their reply to Prentice, suggest that the Baudena et al. (2015) example shows a disadvantage of the remote sensing approach in the present as well. However, the example does not support their case:**

> **• It is the result of a contrived experiment that does not reflect how most global fire-vegetation models actually work.**

This is true. See above.

**• The only global fire-vegetation model I can think of that does directly input satellite-derived burned area (LM3-FINAL.1; Rabin et al., 2018) would not be negatively affected by that input in the present. This is because LM3-FINAL.1 (a) only applies those burned areas on cropland and pasture, thus avoiding the problem with bad fire inputs leading to bad community composition, and (b) uses constant combustion completeness and fractional mortality factors that would not be affected by fire occurring on wet vs. dry days. Rabin et al. (2018) do acknowledge that the use of this input is problematic when applied outside the period of its derivation.**

Yes. I agree that studies of Rabin et al. (2018), Rabin et al. (2015) demonstrate that satellite-derived input for burned areas with some restrictions mentioned by reviewer 2 (listed in (a) and (b)) can be successfully applied in global fire models. Rabin et al. (2018) also demonstrated that relatively small sub-set of satellite-derived input can be successfully used for optimization of parameters of global fire model. Similar method was also applied by Khvostikov et al. (2015) for optimization of parameters of a dynamic global vegetation model for Russia for better description of simulated land cover. I also think that satellite-derived areas burnt should be used for optimization of parameters for SEVER-FIRE in the future, but once more will promote developing at first instance a first-principle global fire model with limited or no satellite derived parameters and only afterwards use satellite-derived data for fine tuning by formal or heuristic optimization. The important findings of Rabin et.al. 2018 and 2015 were presented in the end of Discussion.

**As I've said, I agree with the authors that a first-principles approach could be advantageous because it seems more likely to result in parameterizations that are more robust outside the satellite era, but I cannot think of how any example using historical data would support their case. Instead, I think the best thing the authors could write is what they wrote in their reply to Prentice:**

> **We argue that it would be advantageous if one can produce long-term fire relationships without depending on remote-sensing, which is available for a relatively short period of time (a few decades). Fire return intervals can be of the order of hundreds of years, whereas remote sensing is available for several decades. Therefore using remote sensing to derive relationships implicitly assumes a space for time substitution, which may or may not hold. Also our approach in turn allows the remote sensing to be employed as a valuable evaluation dataset, albeit over this limited time interval.**

Thank you for good advice. We included the paragraph you have mentioned in slightly modified form in Introduction.

**However, I am actually not convinced that SEVER-FIRE even is more grounded in first principles than most other global fire-vegetation models! I see at least one instance where remote sensing or other large-scale, recent historical datasets have been used:**

> **• Equations 1–6, governing lightning ignitions, were derived from national networks of ground-based sensors in the United States and Canada in 1997 (Allen & Pickering, 2002).**

This is true, Allen and Pickering (2002) parametrization is designed from the OTD/LIS observation network for North America. However, the polynomial power four parametrization (number of flashes by convective variables, in our case convective precipitation) is designed based on physical model of induction suggested by Vonnegut (1963). We are now on the way of complete substitution of Allen and Pickering (2002) parametrization by the new entirely physically based model of lightning production (prototype is published in Venevsky (2014))

> **• Equation 9 may also have used such a dataset, although it's not clear exactly how it was parameterized. In their reply to Prentice, the authors mention that the value of $\bar{a}$ for peninsular Spain was derived in the Reg-FIRM description (Venevsky et al., 2002); while I was not able to totally follow the chain of logic presented there, I do understand generally the strategy. However, I do not see the parameterization for the Sahel that, according to the authors' reply to Prentice, is also supposedly in Venevsky et al. (2002). More importantly, even in their reply to Prentice, the authors do not describe what historical fire occurrence data they used to derive Equation 9. Was it satellite data? If so, that undermines the authors' insistence that SEVER-FIRE has an advantage due to independence from parameterizations based on remote sensing data. Or was it instead based on national statistical databases? There are issues with those as well:**

>> **– They only exist in certain wealthy countries.**

>> **– They may not be reliable going back into the mid-20th century.**

>> **– They depend to some extent on the satellite record for recent decades.**

>> **– It would still be basing a part of the model on some external data which, although based on a longer time period than the satellite record, could still fail to be representative of mechanisms far in the past or future.**

We now tried to describe derivation of number of human ignitions in more details. For derivation of equation 9, we used 1) dataset on historical fire

statistics for Spain 1974-1994 (Vazquez and Moreno, 1998); 2) dataset on historical fire statistics for Canada 1965-1991 (Stocks et al., 2002). We admit that these historic datasets do all have four shortcomings listed by Reviewer 2. The statement about Sahel in my reply to I. Colin Prentice is a mistake, what I wanted to say is estimate of $\bar{a}$ for Africa by countries done in Reg-FIRM from (satellite) data of for areas burnt in Africa 1981-1991 (Barbosa et al., 1999). And the changes are shown in the text below Eq. (9).

This is not to say that SEVER-FIRE is an outlier; essentially all global fire-vegetation models are designed to reproduce a limited time series of historical data, either through explicit parameterization processes or through manual model tuning. Global fire models are typically classified into two groupings—purely empirical models and quasi-mechanistic models—which differ in their reliance on parameterizations derived from historical data. See, for example:

> • The anthropogenic ignition components of (most of) the eight models included in Table S1 in the Supplement of Rabin et al. (2017)

> • The parameter estimation (using the Levenberg-Marquardt algorithm) described for the quasi-mechanistic FINAL.1 in Rabin et al. (2018)

> • Purely empirical models such as SIMFIRE (Knorr et al., 2014, 2016)

Thus SEVER-FIRE, rather than being categorically different from most other global fire-vegetation models (a "purely mechanistic" model, perhaps) as Venevsky et al. contend, seems instead to be more first-principles-based only by a matter of degree (i.e., it derives lightning flash rate from weather rather than from a historical-derived climatology, although that derivation does itself depend in part on historical data).

We agree that SEVER-FIRE is a quasi-mechanistic model which is more first-principles-based by a matter of degree, this is now written in Introduction. Clarification for this definition of SEVER-FIRE is done based on terminology of study of Rabin et al. (2017) and Hantson et al. (2016).

Finally, I agree with Reviewer 1 that the satellite record is not unique in its susceptibility to non-representativeness. Even completely accurate, decades-long, ground-based measurements could only be assumed to be representative of the time period covered, with whatever plant species, climate/weather patterns, and anthropogenic activity was there at the time. And of course such records are not completely—or even consistently—accurate anyway! Furthermore, such records are not global in coverage, so even though the problem with space-for-time substitution is lessened relative to the satellite record (not eliminated completely), a space-for-space problem is worsened. Likewise, laboratory-based experiments,

such as those regarding the ignition efficiency of lightning strikes, depend on the species of plant litter involved—even an experiment sampling a wide variety of plant species from across the planet could fail to be representative of species far into the past or future. The brief temporal coverage of the satellite record may make it especially vulnerable to failures of robustness, but other datasets have their own problems.

We certainly agree that both on-ground measurements and laboratory-based experiments are not the Golden Buddha to which we should pray. We now mention characteristic problems related to these kind of data in the Introduction. We just advocate finding understandable explanatory relationships which allow understandable interpretation and visible ways of modification for past and future whether they come from ground, laboratory or satellites or from some theory – does not matter.

Every development team has their own principles that they bring to model construction. If those principles represent a significant break with the dominant mode of thinking in the field, it makes sense to spend time in the model description discussing them. However, Venevsky et al. seem to have a perfectly normal quasi-mechanistic fire model in SEVER-FIRE. Thus, this manuscript should be rewritten to focus on the model itself (especially where it differs from previous models) rather than the philosophy that governed its design.

Thank you for your comments. It is all the question of definition, what is "normal quasi-mechanistic model", what is not. We are, indeed, making an effort to build a first-principle global mechanistic fire model and we are on our way to have it. We have named our model 'Experimental' in order to show that some processes are included in SEVER-FIRE model ad hoc (timing of ignition activity of rural versus urban population, others) as mechanisms are still not described/studied, some processes are simplified (e.g. setting maximum time of fire to two days) and some processes are based on statistical descriptions from satellite data (number of on-ground flashes), as they wait there nearest time to be substituted by mechanistic models. In the Introduction (see below) and further on we followed advice of Reviewer 2 and focused mainly on the SEVER-FIRE model itself and this has been added in the Introduction part, but as well compromised and reserved some place to describe our principles of model design and we hope that the novelty and strategy of SEVER-FIRE can contribute significantly to the field as Reg-FIRM did in its time.

**Other major comments**

**1. Apparent from the comments of Prentice and Reviewer 1, as well as my read of the manuscript, is that the authors need to improve the Introduction, Methods, and Discussion sections to better highlight the**

**novel aspects of SEVER-FIRE.**

Thank you for your kind suggestions. We have re-written the Introduction, clarified the Method, and improve the Discussion, especially more focus on the innovation of SEVER-FIRE. We think that we met this request now.

**2. When explaining novel parts of SEVER-FIRE, the derivation process should always be fully explained—as the authors did for their equations regarding lightning strikes. Such explanation needs to be added for:**

> **• The wealth dependence of anthropogenic ignitions (Eq. 9; as they mention they will do in their reply to Prentice)**

Now explanation is given in the text below Eq. (9), see new text:

*"Equation (9) was obtained using logarithmic regression from geographically distributed observed number of human fires (map of average over 1974-1994 annual number of human fires for Spain (Vazquez and Moreno, 1998), map of average over annual number of human fires by Canadian ecoregions 1961-1995 (Stocks et al., 2002) and map of average over annual number of total fires (assumed to be all human) by African countries 1981-1991 (Barbosa et al., 1999). No division to rural and urban population was assumed when deriving Eq. (9)"*

> **• The limitation of fire duration to two days. This limitation may have contributed to SEVER-FIRE's underestimation of burned area in the boreal region: Korovin (1996) found that almost 70% of the burned forest in Russia over 1947–1992 resulted from fires that burned for more than ten days.**

The limitation of maximum fire duration to two days is explained by range of EFFIS database for Europe. This, of course, limits estimate of areas burnt in remote areas we write in the new text:

*"However, the limitation of maximum fire duration to two days was set due to range in the fire duration of EFFIS database, which covers mainly European domain. Globally this limitation may be not valid for remote high latitude areas, but even in these regions mathematical expectation of fire duration will be close to one day (see Korovin (1996))"*

*In Discussion:*

*"Study and parameterization of fire duration in remote areas is necessary for improvement of area burnt calculation in these areas."*

*In Introduction:*

*"(e.g., setting maximum time of fire to two days but this may be updated and modified in the future by introducing the latest global fire duration datasets (Andela et al., 2018)"*

**3. The factor *timingj*, which modulates the frequency of human ignitions depending on the time of year, seems rather ad-hoc but could nevertheless be of use for many fire models. The authors should demonstrate that including it actually improves the simulation of annual total and/or seasonal timing of burned area.**

We discuss seasonal timing of areas burnt, which are in particularly consequence of *timinig* implementation in paragraph 3.2 of Results. We think, however, that sensitivity and/or optimization study is out of scope for this manuscript and assigned for future work we write in Discussion:

*"In addition, sensitivity study for critical newly implemented features timing and duration and further formal optimization for parameters of SEVER-FIRE model using teaching subset of remote sensing data for observed areas burnt (Khvostikov et al., 2015; Rabin et al., 2018) can further improve performance of the presented global fire model."*

**4. A glaring hole in many global fire models is that they do not allow multi-day burning, and so SEVER-FIRE's inclusion of this is most welcome. However, as with *timingj*, the authors should demonstrate that including this parameterization improves their model.**

We discuss geographical distribution of areas burnt, which are in particularly consequence of duration implementation in paragraph 3.1 of Results. We think, however, that sensitivity and/or optimization study is out of scope for this manuscript and assigned for future work we write in Discussion:

*"In addition, sensitivity study for critical newly implemented features timing and duration and further formal optimization for parameters of SEVER-FIRE model using teaching subset of remote sensing data for observed areas burnt (Khvostikov et al., 2015; Rabin et al., 2018) can further improve performance of the presented global fire model."*

**5. I disagree with Reviewer 1's critique that the paper should be condensed by removing previously-published model components and instead directing readers to those publications. It is too easy to gloss over important differences that may have arisen in the time since the original publication, and makes it too difficult for the reader to learn about the model. One alternative could be to move explanation of non-novel model parts to one or more Appendices (or, less preferably in my opinion, a separate Supplement). The authors should also consider constructing a table-based description of their model to match the form of the**

**supplementary tables in Rabin et al. (2017). This would enable a much simpler comparison between SEVER-FIRE and the models described there, and would ensure a complete description of all relevant aspects of the model.**

Thank you for this comment. This comment is conflict from the Reviewer 1. So, we tried to put as much clarity as possible to our model description now. As for the suggestion to put equations of SEVER-FIRE in Table form of Rabin et al. (2017) we decide not to do it for this version of the model. We are now working on the follow-up version of SEVER-FIRE which is supposed to be closer to a first-principle model. We have an intention to join FireMIP project and run a next version within the FireMIP protocol. After these actions comparison of model structure in tabular form of Rabin et al. 2017 and simultaneous comparison results of SEVER-FIRE and other FireMIP models will have bigger sense and will be more helpful for other fire modelers.

6. **The authors should explain why the model outputs were compared to GFED2, instead of the more recent GFED3(s) or GFED4(s), which would have a number of advantages:**

> **• These datasets cover nearly twice the time period as GFED2, which would increase the time period available for comparison— which the authors acknowledge as a weakness.**
>
> **• GFED3 incorporated an improved burned area detection algorithm (Giglio et al., 2010).**
>
> **• GFED4 incorporated further improvements to the burned area detection algorithm (Giglio et al., 2013).**
>
> **• The "s" versions of GFED3 and GFED4 are boosted by burned area estimated for small fires that the original algorithms fail to detect (Randerson et al., 2012).**

Historically, validation of presented version of SEVER-FIRE model (SEVER-FIRE v1.0) was accomplished just in time when GFED3 version was finalized. As we were building Experimental version of global fire model which was supposed to be further developed we decided not to do re-validation with GFED3. The basis for this decision was that qualitatively and quantitatively GFED2 and GFED3 are similar (see Figure 10 in Giglio et al. (2010)). GFED3 and GFED4 are also very similar (see comparison in Giglio et al. (2013)) and at http://www.globalfiredata.org/figures.html). Versions with small fires have too large uncertainties (van der Werf et al., 2017) (Chuvieco et al., 2016)) which may obscure our goals of analysis of features of fire model. As we plan further development of SEVER-FIRE with joining FireMIP soon we have intention to use GFED4s or GFED4 version for new validation similarly to other FireMIP members. So, we think for the purposes of our current SEVER-FIRE model

status presented validation with GFED2 is sufficient.

**Minor comments and technical corrections**

**1. P10 L24–25: This sentence should cite the "other global fire models,"
as well as perhaps Rabin et al. (2017), which provides a comprehensive
overview and comparison of a number of global fire models**.

Accept and added.

**2. P11 L15 (Eq. 12): This equation structure does not seem to account for
the fact that, for a given rate of linear spread, an older fire has a longer
fireline and thus will add more burned area per unit time than a more
recent fire. This could be a contributing factor to the underestimation of
burned area in boreal regions, where large, long-lasting fires contribute
significantly to total burned area. I do not consider this a critical issue,
but it's something the authors should definitely mention.**

Mentioned now.

**3. P12 L17–26 (Sect. 2.2.2): It would be nice to see, probably in a
Supplement, figures showing the input data described here.**

Thank you for your suggestions. We uploaded the input socio-economic data
to the GutHub together with the code of SEVER-FIRE in case others may need
it. See code availability. And this have been mentioned in section 2.2.

**4. P13 L18: "As a DGVM" should be deleted—there are certainly DGVMs
that have the capability to output results that reflect the vegetated area in
a gridcell. The authors should also explain (a) why they found it necessary
to adjust the GFED data, rather than simply adjusting the SEVER outputs,
and (b) what the net impacts of their adjustments were on global burned
area.**

Accept. Because after re-gridding of GFED dataset to SEVER lan-lat grid
coastlines of both datasets are still different. Impact of this GIS operations on
global burnt are is small (less than 3%). This is in the text now. see section 2.4.

**5. P14 L19–21: A citation of Lasslop et al. (2015) should be made here**.

Accept.

**6. P16 L14–16: This text implies that the overestimation of fire in India
may have something to do with the fact that the model simulates grass
there. In reality, it's probably because of strong fire suppression resulting
from high fractional coverage of cropland.**

This reservation is now mentioned in the section 3.1.

**7. P18 L14: Mention should be made of the fact that these regions were originally created for use with GFED (Giglio et al., 2006).**

Mentioned now.

**8. Work is needed on the Discussion paragraph about anthropogenic impacts on fire (P21 L7–19):**

> **• Pfeiffer et al. (2013) should be mentioned, since they introduce a number of interesting ideas for modeling of human fire use.**

Mentioned now.

> **• "In Africa for example, the combination of a strong seasonal wet-dry climate with regular human ignitions favours high fire incidence." This sentence does not seem to fit with the idea introduced in the previous sentence; namely, that land use and agricultural practices are likely more directly related to fire incidence than wealth in certain regions.**

The sentence is deleted.

Except for the comments from the both reviewers, we also revised the manuscript carefully based on the kind reviewer from Iain Colin Prentice and editors, meanwhile, some typos are changed as well.

**References cited in this response**

Allen, D. J. and Pickering, K. E.: Evaluation of lightning flash rate parameterizations for use in a global chemical transport model, J. Geophys. Res. Atmospheres, 107, ACH 15-11-ACH 15-21, 2002.

Andela, N., Morton, D. C., Giglio, L., Paugam, R., Chen, Y., Hantson, S., van der Werf, G. R., and Randerson, J. T.: The Global Fire Atlas of individual fire size, duration, speed, and direction, Earth Syst. Sci. Data Discuss., 2018, 1-28, 2018.

Barbosa, P. M., Stroppiana, D., Grégoire, J.-M., and Cardoso Pereira, J. M.: An assessment of vegetation fire in Africa (1981–1991): Burned areas, burned biomass, and atmospheric emissions, Global Biogeochem. Cycles, 13, 933-950, 1999.

Baudena, M., Dekker, S. C., van Bodegom, P. M., Cuesta, B., Higgins, S. I., Lehsten, V., Reick, C. H., Rietkerk, M., Scheiter, S., Yin, Z., Zavala, M. A., and

Brovkin, V.: Forests, savannas, and grasslands: bridging the knowledge gap between ecology and Dynamic Global Vegetation Models, Biogeosciences, 12, 1833-1848, 2015.

Chuvieco, E., Yue, C., Heil, A., Mouillot, F., Alonso-Canas, I., Padilla, M., Pereira, J. M., Oom, D., and Tansey, K.: A new global burned area product for climate assessment of fire impacts, Global Ecol. Biogeogr., 25, 619-629, 2016.

Giglio, L., Randerson, J. T., van der Werf, G. R., Kasibhatla, P. S., Collatz, G. J., Morton, D. C., and DeFries, R. S.: Assessing variability and long-term trends in burned area by merging multiple satellite fire products, Biogeosciences, 7, 1171-1186, 2010.

Giglio, L., Randerson, J. T., van der Werf, G. R., Randerson, J. T., Chen, Y., van der Werf, G. R., Rogers, B. M., and Morton, D. C.: Analysis of daily, monthly, and annual burned area using the fourth-generation global fire emissions database (GFED4)

Global burned area and biomass burning emissions from small fires, J. Geophys. Res. Biogeosciences, 118, 317-328, 2013.

Hantson, S., Arneth, A., Harrison, S. P., Kelley, D. I., Prentice, I. C., Rabin, S. S., Archibald, S., Mouillot, F., Arnold, S. R., Artaxo, P., Bachelet, D., Ciais, P., Forrest, M., Friedlingstein, P., Hickler, T., Kaplan, J. O., Kloster, S., Knorr, W., Lasslop, G., Li, F., Mangeon, S., Melton, J. R., Meyn, A., Sitch, S., Spessa, A., van der Werf, G. R., Voulgarakis, A., and Yue, C.: The status and challenge of global fire modelling, Biogeosciences, 13, 3359-3375, 2016.

Khvostikov, S., Venevsky, S., and Bartalev, S.: Regional adaptation of a dynamic global vegetation model using a remote sensing data derived land cover map of Russia, Environmental Research Letters, 10, 9, 2015.

Korovin, G. N.: Analysis of the Distribution of Forest Fires in Russia. In: Fire in Ecosystems of Boreal Eurasia, Goldammer, J. G. and Furyaev, V. V. (Eds.), 48, Springer, Dordrecht, 1996.

Rabin, S. S., Magi, B. I., Shevliakova, E., and Pacala, S. W.: Quantifying regional, time-varying effects of cropland and pasture on vegetation fire, Biogeosciences, 12, 6591-6604, 2015.

Rabin, S. S., Melton, J. R., Lasslop, G., Bachelet, D., Forrest, M., Hantson, S., Kaplan, J. O., Li, F., Mangeon, S., Ward, D. S., Yue, C., Arora, V. K., Hickler, T., Kloster, S., Knorr, W., Nieradzik, L., Spessa, A., Folberth, G. A., Sheehan, T., Voulgarakis, A., Kelley, D. I., Prentice, I. C., Sitch, S., Harrison, S., and Arneth, A.: The Fire Modeling Intercomparison Project (FireMIP), phase 1:

experimental and analytical protocols with detailed model descriptions, Geosci. Model Dev., 10, 1175-1197, 2017.

Rabin, S. S., Ward, D. S., Malyshev, S. L., Magi, B. I., Shevliakova, E., and Pacala, S. W.: A fire model with distinct crop, pasture, and non-agricultural burning: use of new data and a model-fitting algorithm for FINAL.1, Geosci. Model Dev., 11, 815-842, 2018.

Stocks, B. J., Mason, J. A., Todd, J. B., Bosch, E. M., Wotton, B. M., Amiro, B. D., Flannigan, M. D., Hirsch, K. G., Logan, K. A., Martell, D. L., and Skinner, W. R.: Large forest fires in Canada, 1959–1997, Journal of Geophysical Research: Atmospheres, 107, FFR 5-1-FFR 5-12, 2002.

van der Werf, G. R., Randerson, J. T., Giglio, L., van Leeuwen, T. T., Chen, Y., Rogers, B. M., Mu, M., van Marle, M. J. E., Morton, D. C., Collatz, G. J., Yokelson, R. J., and Kasibhatla, P. S.: Global fire emissions estimates during 1997–2016, Earth System Science Data, 9, 697-720, 2017.

[revised manuscript text omitted]